# DYNA-THINK: SYNERGIZING REASONING, ACTING, AND WORLD MODEL SIMULATION IN AI AGENTS

## ABSTRACT

Recent progress in reasoning with large language models (LLMs), such as DeepSeek-R1, demonstrates impressive capabilities in domains like mathematics and coding, by exhibiting complex cognitive behaviors such as verification, goal decomposition, and self-reflection. However, it is unclear what behavior is effective and what behavior is missing for agentic tasks. In this work, we propose Dyna-Think, a thinking framework that integrates *planning with an internal world model* with reasoning and acting to enhance AI agent performance. To enable Dyna-Think, we propose Dyna-Think Imitation Learning (DIT) and Dyna-Think Dyna Training (DDT). To initialize a policy with Dyna-Think, DIT reconstructs the thinking process of R1 to focus on performing world model simulation relevant to the proposed (and planned) action, and trains the policy using this reconstructed data. To enhance Dyna-Think, DDT uses a two-stage training process to first improve the agent's world modeling ability for next state prediction and critique generation, and then improve the agent's action via policy training. We evaluate our methods on OSWorld and WindowsAgentArena, and demonstrate that Dyna-Think improves the agent's in-domain and out-of-domain performance, achieving similar best-of-n performance compared to R1 while generating 2x less tokens on average. Our extensive empirical studies reveal that 1) using critique generation for world model training is effective to improving policy performance; and 2) improving a world model is effective to improving the performance of its AI agent. Our results suggest a promising research direction to integrate world-model simulation into AI agents to enhance their reasoning, planning, and acting capabilities.

## 1 INTRODUCTION

Autonomous AI agents powered by large language models (LLMs) have offered substantial promise in real-world applications, automating digital tasks such as software engineering (Jimenez et al., 2024; Wang et al., 2024b; Yang et al., 2024a), web navigation (Liu et al., 2018; Yao et al., 2023a; Zhou et al., 2024b; Koh et al., 2024a), computer-use (Xie et al., 2024; Anthropic, 2025b; Qin et al., 2025), and mobile device control (Rawles et al., 2023; 2024; Trivedi et al., 2024). As many computer use tasks require agents to interact with complex environments to achieve long-term objectives, one critical challenge is to reason and act efficiently over a large decision space.

Recent methods of test-time scaling (Snell et al., 2024) offer a potentially promising solution. For example, Yu et al. (2023; 2024b); Zhou et al. (2024a) show that LLM agents can significantly improve performance using search algorithms such as Monte Carlo Tree Search (MCTS) to allow additional interactions with the environment before decision-making. However, these methods require a large amount of expensive, time-consuming interactions with a real (or a separately learned) world model, limiting their applicability in real-world scenarios. Alternatively, many recent work (OpenAI, 2024; Valmeekam et al., 2024; Zhou et al., 2025) finds that LLM agents can also improve their performance by "thinking" longer, effectively internalizing parts of search into their reasoning process. For example, models such as OpenAI o1/o3 (OpenAI, 2024; 2025a), Claude-3.7-Sonnet (Anthropic, 2025a), and DeepSeek-R1 (DeepSeek-AI et al., 2025) generate significantly longer reasoning chains when faced with challenging tasks, exhibiting behaviors such as self-reflection, goal decomposition, verification, exploration, and more (Gandhi et al., 2025). Although many of these behaviors are crucial for solving challenging long-horizon tasks, recent work also finds that these models suffer from problems such as overthinking and fact-ignoring (Cuadron et al., 2025; Zhou et al., 2025), and

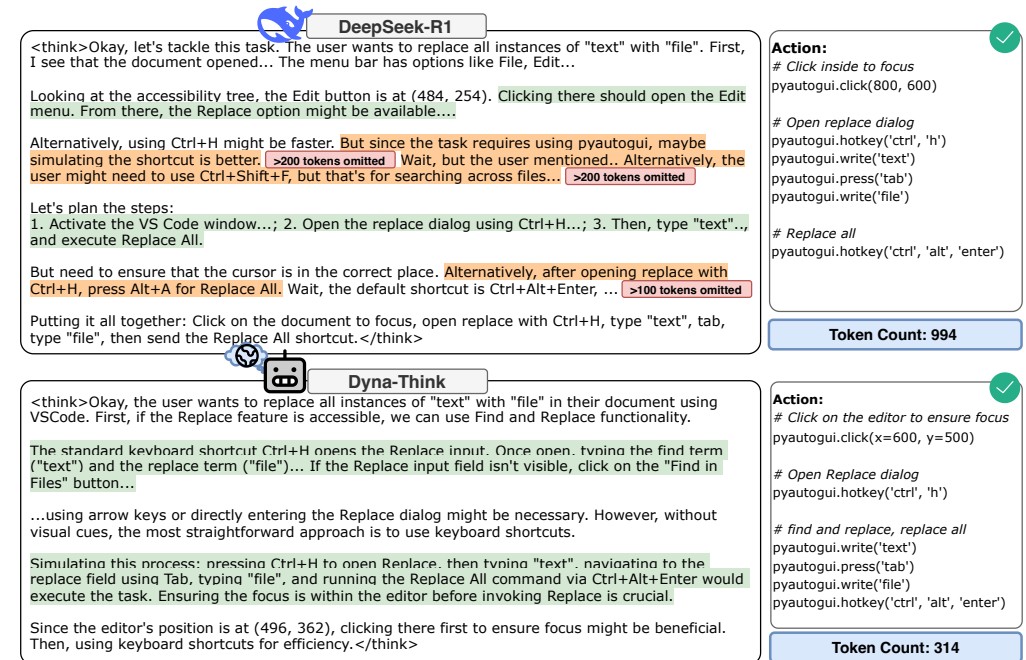

Figure 1: Dyna-Think focus on integrating world model simulation (shown in green) with reasoning and acting. Behaviors that is not necessary to cause/has unclear contribution to the final action is shown in orange. After training, we find Dyna-Think achieves similar BoN performance compared to R1, while generating 2x less tokens on average and being only 32B in size.

are less efficient in certain tasks compared to non-reasoning models such as GPT-4o (Zhou et al., 2025). It is thus unclear *what* type of "thinking" is crucial to an agent's performance for long-horizon tasks, and *how* to improve it via learning from experiences.

The successes in Go, games, and robotics reveal that modeling and interacting with the environment is the key to solving challenging long-horizon tasks. The Dyna algorithms (Sutton, 1991) offer a promising approach to combine real-world interactions with planning via simulation to improve an agent's policy. However, modeling the whole environment for tasks such as computer-use (e.g., predicting screen content after entering 'amazon.com' in chrome) is challenging (Pathak et al., 2017; Wang et al., 2024a; Fang et al., 2025; Gu et al., 2025). In contrast, research in cognitive sciences (Marr, 1982; Rao & Ballard, 1999) shows that the human brain encodes a *compressed* representation of the external world, capturing only statistical regularities and meaningful structures related to current tasks. We thus propose Dyna-Think, a thinking framework that performs *"compressed" world model simulation* based on model-generated states and critiques (rewards), and *integrates it with reasoning and acting* to improve an AI agent's performance. Figure 1 illustrates the difference between the Dyna-Think agent and DeepSeek-R1 using an example.

The Dyna-Think framework consists of Dyna-Think Imitation Learning (DIT) and Dyna-Think Dyna Training (DDT). DIT improves a baseline agent policy by first distilling simulated experiences from R1-generated thinking tokens, and then revising the baseline policy using the simulated experiences via supervised learning. We finds models after DIT achieve a similar performance to R1-distilled models, but with 2X less thinking tokens on average. To further improve DIT, we propose DDT, is an extension to the Dyna-Q method (Sutton & Barto, 2018) that combines policy and world model learning during online training. Similar to Dyna-Q, DDT first constructs policy and world model training data using online rollouts in a real (digital) environment. Different from Dyna-Q, DDT is applicable to the problems with arbitrarily large state-action space by leveraging the prior knowledge encoded in a pre-trained LLM, and performs both policy learning and world model training based on *a single LLM*. During world model training, the model encodes state transitions by optimizing training objectives of next-state prediction and critiquing the states generated by its own. During policy learning, the agent's actions are improved using successful rollouts via reinforcement learning.

We evaluate Dyna-Think on OSWorld (Xie et al., 2024) and WindowsAgentArena (Bonatti et al., 2024). Results show that Dyna-Think improves the agent's in-domain and out-of-domain performance compared to only performing policy training (e.g., reinforcement learning) or training a separate world model (e.g., as in Dyna-Q), and that Dyna-Think leads to highly capable and cost-effective agents. For example, our 32B-parameter Dyna-Think model achieves a similar best-of-n performance compared to the much larger 685B-parameter R1 model, with 2x less tokens on average. Detailed analysis also reveals that 1) the critique-style world model training is effective in improving policy; and 2) AI agents with a stronger world model achieve better performance. Thus, our work demonstrates the potential of integrating planning and learning to develop future agents powered by reasoning models.

## 2 RELATED WORK

**Computer-Use Agents**   Computer-use agents powered by large (multimodal) language models aim to automate tasks by controlling a computer, typically via GUI interactions with a virtual Ubuntu/Windows (Xie et al., 2024; Bonatti et al., 2024; Anthropic, 2025b). Early computer-use methods include reactive agents (Yao et al., 2023b; Xie et al., 2024) that directly prompt an LLM (e.g., GPT-4o) to make decisions on immediate observations without simulation or planning approaches. Recent advances include: 1) search-based methods (Zhou et al., 2024a; Koh et al., 2024b; Yu et al., 2024b) that augment LLMs with look-ahead search algorithms such as MCTS; and 2) hierarchical-planning methods that orchestrate multiple modules, tools, and LLMs to complete a task (Agashe et al., 2024; 2025; Liu et al., 2025; Gou et al., 2025; Yang et al., 2024b). However, search-based methods significantly increase inference time due to using additional interactions with the external environment; and hierarchical-planning methods often require complex human-designed rules and heuristics to coordinate multiple modules. We introduce our Dyna-Think framework to enhance the thinking process of a single LLM agent by performing action-centric world model simulation.

**Training AI Agents**   In addition to improving agent's performance at test-time, many works also explored methods to improve performance via training. Recent methods include Chen et al. (2023); Zhang et al. (2024); Zeng et al. (2023); Lai et al. (2024); Xu et al. (2025) which perform supervised training using human or machine generated trajectories based on direct prompting; Qin et al. (2025); Su et al. (2025); Hong et al. (2024) which improves an agent's ability such as GUI grounding and error recovery via SFT/DPO on human-machine collaboration data; and Bai et al. (2024); Chen et al. (2025); Jin et al. (2025), which explores using direct RL to improve an agent's policy within complex tool-use/android-based environments. These methods focus on policy improvements based on an existing thinking paradigm (e.g., ReACT or R1-style thinking). We propose DIT to improve the agent's reasoning ability by integrating world model simulation into its thinking process, and DDT to further enhance policy training via world model training.

**World Models**   Obtaining real-world data for large-scale training or test-time search is expensive and may cause unintended consequences. To this end, early methods such as Peng et al. (2018); Wu et al. (2018); Fang et al. (2025) consider Dyna-style training that separately trains a world model and then enhance policy training using synthetic rollouts; and prompting LLMs as world models Hao et al. (2023); Kim et al. (2024) in simplified environments such as BlocksWorld (Valmeekam et al., 2023). Recently, Chae et al. (2025); Gu et al. (2025) trains a world model using a large corpus of web data to facilitate inference-time algorithms such as MCTS. To our knowledge, this is the first work to propose internalizing world model simulation into the agent's thinking process, and to introduce methods to further improve the agent's policy by both world model and policy training.

**Dyna Algorithms**   Dyna algorithms (Sutton, 1991) combine model-based and model-free methods to learn optimal policies. These methods improve the training efficiency of $\pi(\theta)$ by combining real-world interaction with simulated planning. Given a set of real-world rollout data, Dyna algorithms typically 1) separately trains a world model $\mathcal{W}(\mu)$ using these rollouts; 2) perform additional rollouts using $\mathcal{W}(\mu)$; and 3) trains $\pi(\theta)$ using both the real-world and simulated rollouts. Given a large set of training tasks to perform rollouts, this two-stage training process can be repeated multiple times. Applications of Dyna algorithms with language models include Deep Dyna-Q (Peng et al., 2018), Switch-DDQ (Wu et al., 2018), Pseudo-DDQ (Zou et al., 2020) and more, covering domains such as movie-ticket booking and e-commerce product recommendations.

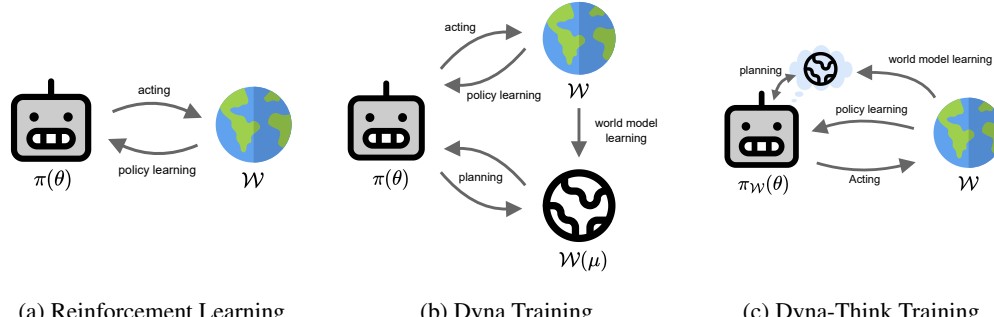

(a) Reinforcement Learning      (b) Dyna Training      (c) Dyna-Think Training

Figure 2: Our Dyna-Think framework synergizes planning with world model simulation in an agent's reasoning process, and performs both world model training and policy training with $\pi_\mathcal{W}(\theta)$.

## 3 DYNA-THINK FRAMEWORK

Accurate world simulation is challenging for AI agents. We propose Dyna-Think, a two-stage training method to address this. First, we introduce DIT (Section 3.2) to synergize reasoning, acting, and world model simulation in an agent's thinking process via imitation learning. Then, we introduce DDT (Section 3.3) to further improve its policy and world modeling ability via Dyna-style training.

### 3.1 TASK DEFINITION

Completing tasks in a complex environment (e.g., with a computer) is typically formulated as a partially observable Markov decision process (POMDP) of $(\mathcal{S}, \mathcal{O}, \mathcal{A}, \mathcal{T}, \mathcal{R})$. In the context of computer-use, $\mathcal{S}$ represents the set of possible computer states, $\mathcal{O}$ is the set of observations available to the agent, $\mathcal{A}$ represents the set of executable actions as left/right-clicks at a location, typing, and pressing keyboard shortcuts, $\mathcal{T}$ is the transition function $\mathcal{T} : \mathcal{S} \times \mathcal{A} \rightarrow \mathcal{S}$ that determines the next state given the current state and action, and $\mathcal{R}$ is the reward function that provides feedback to the agent based on its actions. In typical computer-use benchmarks, $o \in \mathcal{O}$ is either a screenshot of the current computer screen or its text-based representation (e.g., accessibility tree); and $r = 0$ is zero if the task has not terminated, and $r_T = \{0, 1\}$ if the agent failed/succeeded the task at the end. Since $|\mathcal{A}|$ is extremely large and tasks such as computer-use often requires many steps to complete, many agent benchmarks remain challenging even for current state-of-the-art LLMs.

Given a task, a computer-use agent iteratively interact with the environment by generating an action $a_i$ based on the current observation $o_i$ and the past history of observations and actions $(o_{<i}, a_{<i})$. In this work, we refer to such language model policies parametrized by $\theta$ as $\pi(\theta)$, and learnt world models parametrized by $\mu$ as $\mathcal{W}(\mu)$. We denote policies that performs interaction with an internal world model during its thinking process as $\pi_\mathcal{W}(\theta)$.

### 3.2 DYNA-THINK IMITATION LEARNING

Many expert LLMs capable of extensive thinking (e.g., DeepSeek R1, Claude-3.7-Sonnet, and OpenAI's o3) exhibit complex behaviors during its thinking process. This includes being able to perform verification, self-reflection, error recovery, world modeling, and more. However, it is unclear which behaviors are critical for decision-making in long-horizon AI agent tasks. In our preliminary study, we find training on long CoT data with simulations/knowledge unrelated to the final action degrades performance, a phenomenon also found in other domains (Zhou et al., 2024c; Wu et al., 2025). To enable Dyna-Think for "weak" non-reasoning models, we propose **Dyna-Think Imitation Learning** (DIT) to first construct reasoning data that emphasizes on world simulation and perform imitation learning. Specifically, DIT reconstructs R1's thinking process to only contain text related to reasoning, the final action, and the *world modeling simulation* related to the final action; and then trains a policy using the reconstructed data. To perform this reconstruction, we few-shot prompt GPT-4o (see Appendix D for examples and details on using other reconstruction prompts). After training, we refer to these models as $\pi_\mathcal{W}(\theta)$, highlighting their ability to perform world-model simulations during thinking.

### 3.3 DYNA-THINK DYNA TRAINING

Despite DIT learning, $\pi_{\mathcal{W}}(\theta)$ at inference time can still make mistakes when facing unseen states unseen from training. However, creating a DIT training set that covers all possible states $|\mathcal{S}|$ is intractable. To further improve DIT-trained models, we propose **Dyna-Think Dyna Training** (DDT), a Dyna-style method that performs both policy and world model learning during online training. Similar to Dyna, we first collect policy and world model learning data by performing $\pi_{\mathcal{W}}(\theta)$ rollouts in the real environment. Different from Dyna, we then directly perform both policy and world model learning on the same $\pi_{\mathcal{W}}(\theta)$ model. We illustrate our method in Figure 2.

**Policy Training**  Alike reinforcement learning, the policy training stage aims to directly improve a policy using environment feedback. This is achieved by first generating online rollouts with $\mathcal{W}$ to collect a trajectory $\tau = (o_0, a_1, o_1, a_2, ..., a_T)$, and then constructing a policy dataset that trains $\pi_{\mathcal{W}}(\theta)$ to predict each action $a_i$ given the previous context context $(o_0, a_{<i}, o_{<i})$ based on a reward function (e.g., task success). Optimizing such reward can be done using algorithms such as PPO (Schulman et al., 2017), GRPO (Shao et al., 2024), DPO (Rafailov et al., 2024), or Rejection Sampling (Bai et al., 2022; Touvron et al., 2023).

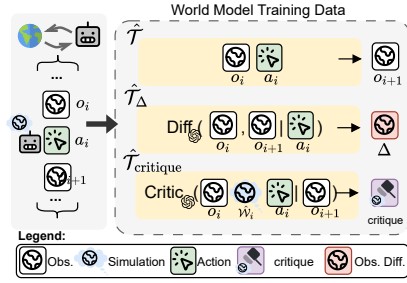

**World Model Training**  Given a rollout trajectory $\tau$, we also construct world model training dataset to train $\pi_{\mathcal{W}}(\theta)$ to model a *function* of the environment transition $\hat{\mathcal{T}}_f(o_i, a_i)$. We experiment with three different functions for $\hat{\mathcal{T}}_f$ in this work. $\hat{\mathcal{T}}(o_i, a_i) \to o_{i+1}$ which directly models the next state; $\hat{\mathcal{T}}_\Delta(o_i, a_i) \to \Delta(o_i, o_{i+1}|a_i)$ which trains to predict relevant *changes* in the next state caused by $a_i$; and $\hat{\mathcal{T}}_{\text{critic}}(o_i, a_i) \to \text{critic}(o_i, a_i|o_{i+1})$ which trains to predict a *critique* generated by an LLM (GPT-4o) by comparing the

Figure 3: Three different forms of world model data experimented in DDT.

world model simulations in $a_i$ with the previous and next state. An example world model simulation could be "After opening the terminal with Ctrl+Alt+T, type 'cp dir2/hi.txt dir1/' to copy ...", and an example critique is "Wait, maybe we need to first ensure that 'dir1' exists in the current directory...". We illustrate these three functions in Figure 3. To utilize these objectives to *enhance the world modeling ability of the policy model*, we optimize next-state prediction $\hat{\mathcal{T}}$ and state-changes prediction $\hat{\mathcal{T}}_\Delta$ as an "auxiliary" task trained by standard language modeling loss alongside policy training; and we optimize $\hat{\mathcal{T}}_{\text{critic}}$ by injecting the generated critique back into the action ($a_i' = a_i \oplus \text{critic}$) and then train the policy to predict critic tokens in $a_i'$ using language modeling loss. Intuitively, $\hat{\mathcal{T}}$ and $\hat{\mathcal{T}}_\Delta$ enhance a policy's world modeling ability implicitly, while $\hat{\mathcal{T}}_{\text{critic}}$ is more explicit. For more details, please see Appendices E.1 and E.2.

Finally, we combine policy and world model training, DDT follows Dyna methods and perform two-stage learning by first training $\pi_{\mathcal{W}}(\theta)$ on the world model dataset, and then training $\pi_{\mathcal{W}}(\theta)$ on the policy dataset. For pseudocode, please see Algorithm 1. For other implementation details such as critique prompts and how training data is formatted, please refer to Appendix E.

## 4 EXPERIMENTS

### 4.1 BENCHMARKS

We mainly evaluate our methods on OSWorld (Xie et al., 2024), a diverse benchmark consisting of 369 open-ended computer tasks that involves real web and desktop apps in open domains, OS file I/O, and workflow tasks. Tasks are categorized into 10 different domains based on different desktop applications involved, such as OS Terminal, LibreOffice Calc, LibreOffice Impress, LibreOffice Writer, Chrome, VLC Player, Thunderbird, VS Code, GIMP and Workflow.

To evaluate the self-improving ability of AI agents, many methods (Huang et al., 2022; Yu et al., 2024a; Chen et al., 2025; Jin et al., 2025) consider domains where the model has non-trivial initial performance. However, in OSWorld we find many domains such as LibreOffice Calc and Workflow

to be extremely challenging, with current models achieving less than 10% success rate or solving only 2-3 tasks often due to state representation issues (see Appendix F.1). To this end, we evaluate our method on 5 domains that are more accessible to existing models, including OS, Chrome, VS Code, GIMP, and Thunderbird; and also *additionally on WindowsAgentArena* (Bonatti et al., 2024) to further measure model's generalization ability to a different operating system (Windows OS).

## 4.2 EXPERIMENTAL SETUP

**Training/Testing Dataset**    Since most computer-use benchmarks were not designed for training, the number of tasks available in each domain is often limited. We thus construct a training and test set by 1) manually augmenting existing tasks in each domain to increase its size; 2) construct training/test splits for each domain, and 3) held out two domains (GIMP and Thunderbird) from training to separately measure **In-domain (ID)** and **Out-of-domain (OOD)** performance. To augment a task, we follow the principle that 1) action sequence that correctly completes task $A$ does *not* complete the augmented task $A'$; and 2) the augmented task $A'$ can still be evaluated using OSWorld's evaluation scripts after some adjustments. Please refer to Appendix F.2 for more details and example augmented tasks. In Table A2 we report the training and testing dataset statistics. For WindowsAgentArena, we directly test on the official tasks as only OSWorld tasks were used in training.

**Evaluation Details**    We evaluate all runs using the accessibility-tree mode (i.e., text-only) on both benchmarks, and report task success for ID and OOD tasks. We use the same inference prompts and hyperparameters (e.g., temperature of 1.0) provided by OSWorld, but extend the maximum number of steps per task to 30 in order to measure scaling abilities. To provide a more robust evaluation for these long-horizon tasks, we report the average success rate (Avg) and the Best-of-N success rate (BoN) over 3 runs. Each evaluation is ran with 3 threads, lasting on average 18-24 hours to complete.

**Training Details**    We train all of our models based on Qwen2.5-32B-Instruct (Qwen et al., 2025) using 8xH100 GPUs. We use 32B models due to the limited learning ability of smaller models for complex thinking patterns (Li et al., 2025). For simplicity, we use rejection sampling as the optimization algorithm for policy training, and SFT for world model training. For all runs, we use AdamW with a linear learning rate scheduler with 10% warmup steps. In the two-stage Dyna-Think Dyna Training, we first perform world model training for 2 epoch with a max learning rate of 5e-6, and then continue training on policy learning data for 3 epochs with a max learning rate of 5e-6.

## 4.3 MAIN RESULTS

**Baselines**    We compare our Dyna-Think framework against 1) training-free methods based on expert LLMs; and 2) related training methods. For training-free methods, we consider prompting LLMs such as o3-mini and DeepSeek-R1 using REACT. For training-based methods, we consider 1) **Reinforcement Finetuning** (RFT) which only peforms policy learning, by finetuning on correct rollouts after rejection sampling (Best-of-3); and 2) the vanilla **Dyna** algorithm (we follow Fang et al. (2025)) which performs world model learning with a separate language model $\mathcal{W}(\mu)$, and then trains the policy using correct rollouts[1] collected with *both* the real $\mathcal{W}$ and the learned $\mathcal{W}(\mu)$. Our DDT method only uses rollouts with $\mathcal{W}$ and performs policy and world model learning on a single model.

For a fair comparison, we use the same set of rollout trajectories (116 in total) to construct world model (116) and policy training data (35 after rejection sampling, a similar data size as Yu et al. (2024b)) for all training methods, whenever possible.

**Results**    We report our results on OSWorld in Table 1, and results on WindowsAgentArena on Table 2. We first compare training-based methods. In Table 1, we find that DDT training, especially when trained with next-state prediction (DDT($\hat{\mathcal{T}}$)) and critique prediction (DDT($\hat{\mathcal{T}}_{\text{critic}}$)) improves upon both RFT and vanilla Dyna in both average success rate and BoN success rate. This indicates that 1) world model training benefits policy training; and 2) directly training $\pi_{\mathcal{W}}(\theta)$ as a world model is more effective than planning with a separately trained $\mathcal{W}(\mu)$. We believe this is because vanilla Dyna requires predicting the *full* next screen state at every step to generate synthetic trajectories for

---

[1]When rolling out with the learned $\mathcal{W}(\mu)$, correctness of a trajectory is evaluated using an LLM (Fang et al., 2025). We use GPT-4o in this work.

Table 1: Average (AVG) and Best-of-N (BoN) success rate on OSWorld after policy and world model learning. $|\pi|$ and $|\mathcal{W}|$ denotes the number of trajectories used during policy learning and world model learning, respectively. "Gen. Token" denotes the 10th and 90th percentiles of output token lengths per prompt generated by the models. All training are based on Qwen-2.5-32B-Instruct. All methods are evaluated over 3 runs. For overall performance (All), we report average success $\pm$ standard deviation.

| Method | $|\pi|$ | $|\mathcal{W}|$ | Gen. Token (10%-90%) | Avg Success Rate | | | BoN Success Rate | | |
|---|---|---|---|---|---|---|---|---|---|
| | | | | All(174) | ID(123) | OOD(51) | All(174) | ID(123) | OOD(51) |
| - (GPT-4o-2024-11-20) | - | - | 1.0x | 16.5±2.5 | 19.5 | 9.2 | 31.6 | 36.6 | 19.6 |
| - (Qwen2.5-32B-Instruct) | - | - | 0.2x-0.8x | 14.2±2.2 | 18.9 | 2.5 | 27.0 | 35.0 | 7.8 |
| - (R1-distill-70B) | - | - | 0.6x-2.2x | 7.6±2.0 | 9.8 | 2.5 | 15.5 | 20.3 | 3.9 |
| - (o3-mini-2025-01-31) | - | - | 3.4x-7.0x | 20.5±1.4 | 23.3 | 13.7 | 31.0 | 36.6 | 17.6 |
| - (R1) | - | - | 1.7x-4.9x | 31.2±1.4 | 35.8 | 20.2 | 44.8 | 48.8 | 35.3 |
| DIT(R1) | - | - | 1.1x-2.5x | 22.6±1.0 | 26.8 | 12.4 | 35.6 | 40.7 | 23.5 |
| +RFT | 35 | - | 1.0x-2.7x | 23.2±1.0 | 26.0 | 16.3 | 38.5 | 43.1 | 27.5 |
| +vanilla Dyna | 94 | 116 | 1.1x-2.5x | 24.1±2.1 | 26.8 | 17.6 | 35.6 | 29.3 | 31.4 |
| +DDT($\hat{\mathcal{T}}$) | 35 | 116 | 1.1x-2.7x | 25.9±1.4 | 28.7 | **19.0** | 43.1 | 47.2 | **33.3** |
| +DDT($\hat{\mathcal{T}}_\Delta$) | 35 | 116 | 1.2x-3.5x | 23.2±0.7 | 27.6 | 12.4 | 38.5 | 44.7 | 23.5 |
| +DDT($\hat{\mathcal{T}}_{\text{critic}}$) | 35 | 116 | 1.2x-2.7x | **26.6**±1.5 | **30.3** | 17.6 | **44.3** | **49.6** | 31.4 |

Table 3: Integrating world model simulation (WM Sim) into reasoning. All training are based on Qwen2.5-32B-Instruct. "R1 no-think" refers to only training on tokens *after* the "</think>" tag. Since o3-mini API does not return the model's thinking process, it is unclear if it performs world modeling.

| Method | WM Sim? | Gen. Token (10%-90%) | Avg Success Rate | | | BoN Success Rate | | |
|---|---|---|---|---|---|---|---|---|
| | | | All(174) | ID(123) | OOD(51) | All(174) | ID(123) | OOD(51) |
| - (GPT-4o-2024-11-20) | ✗ | 1.0x | 16.5±2.5 | 19.5 | 9.2 | 31.6 | 36.6 | 19.6 |
| - (Qwen2.5-32B-Instruct) | ✗ | 0.2x-0.8x | 14.2±2.2 | 18.9 | 2.5 | 27.0 | 35.0 | 7.8 |
| - (R1) | ✓ | 1.7x-4.9x | 31.2±1.4 | 35.8 | 20.2 | 44.8 | 48.8 | 35.3 |
| DIRECT DISTILL(4o) | ✗ | 0.5x-1.7x | 15.3±2.4 | 18.5 | 7.8 | 28.7 | 34.1 | 15.7 |
| DIRECT DISTILL(R1 no-think) | ✗ | 0.1x-0.5x | 17.1±2.1 | 19.3 | 11.8 | 28.2 | 30.9 | 21.6 |
| DIRECT DISTILL(R1) | ✓ | 1.6x-6.0x | 20.9±1.0 | 24.6 | 11.8 | **36.2** | **42.3** | 21.6 |
| DIT(R1) | ✓ | 1.1x-2.5x | **22.6**±1.0 | **26.8** | **12.4** | 35.6 | 40.7 | **23.5** |

training. This is an extremely difficult task in complex, long-horizon environments like OSWorld. In contrast, DDT avoids full state prediction and instead trains the model to produce *(partial) textual descriptions of future states within its reasoning process* (see the lower part of Figure 1 as an example), which is more tractable. Next, we find that training with critique data showed strong performance compared to training on state-difference (DDT($\hat{\mathcal{T}}_\Delta$)) and on next-state prediction (DDT($\hat{\mathcal{T}}$)). We believe this is because these critique data provide a more direct signal for $\pi_{\mathcal{W}}(\theta)$ to improve its world modeling and planning ability *during inference* (see Section 5.3 for more quantitative study).

Moreover, we find that training with the next-state prediction objective (DDT($\hat{\mathcal{T}}$)) shows strong performance for OOD tasks, especially on WindowsAgentArena. We believe this is because OOD environments place a greater emphasis on correctly understanding the current state, which is needed for effective planning. The next-state prediction objective in DDT($\hat{\mathcal{T}}$) is thus more effective for this scenario, especially when the model's base capability is not strong. Finally, compared to best training-free methods such as prompting R1(685B), we find Dyna-Think achieves similar BoN score, while generating 2x less tokens on average and being only

Table 2: WindowsAgentArena results.

| Method | Avg Success Rate |
|---|---|
| - (Qwen-32B) | 23.9±1.1 |
| - (R1) | 26.9±1.5 |
| DIT(R1) | 26.9±1.3 |
| + RFT | 28.4±1.8 |
| + vanilla Dyna | 20.9±1.1 |
| +DDT($\hat{\mathcal{T}}$) | **34.9**±1.4 |
| +DDT($\hat{\mathcal{T}}_{\text{critic}}$) | 32.8±1.4 |

32B in size. These results indicate the many tokens/behaviors during R1-style reasoning may not be necessary, and that *focusing on/improving simulation ability* is effective at improving agent performance.

## 4.4 THINKING BEHAVIOR ANALYSIS

We now investigate what behavior is essential for long-horizon AI agent tasks. We compares agents with no-thinking; R1-style thinking, and Dyna-Think. For **no-thinking**, we consider 1) DIRECT DISTILL(4o) which distills from GPT-4o that, to our observation, does *not* perform world modeling

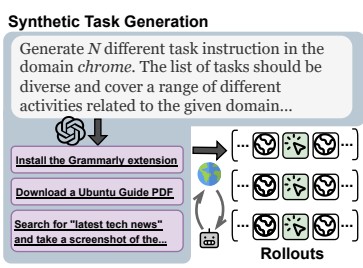

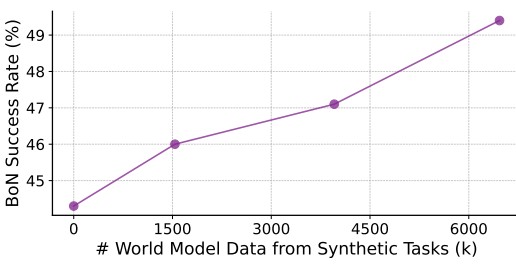

(a) Generating synthetic tasks for rollouts.    (b) Test performance.

Figure 4: Scaling world model learning with synthetic tasks generated by GPT-4o. We use DDT($\hat{\mathcal{T}}_{\text{critic}}$) and train from our best model in Table 3. After world model training, we perform one-round of policy training using a fixed set of policy training data (from Section 4.3).

(Chae et al., 2025); and 2) DIRECT DISTILL(R1 no-think) which trains only on tokens after the thinking process (by removing all text within the '<think></think>' tags in each response). For **R1-style** thinking, we consider DIRECT DISTILL(R1), which trains on the entire thinking process generated by R1. For **Dyna-Think**, we consider DIT(R1). For a fair comparison across different methods, we use the *same set of correct trajectories* obtained by best-of-3 rejection sampling.

We present the results in Table 3. First, we find that DIRECT DISTILL(R1 no-think) deteriorates significantly compared to DIRECT DISTILL(R1), even though they are trained on the same set of trajectories. This indicates that including long-CoT thinking during policy training is beneficial. Next, we find that DIT(R1) achieved similar performance compared to DIRECT DISTILL(R1), despite generating 2x less tokens on average. This suggests that the ability to perform *reasoning with world-model simulations* is the central part in R1-style thinking, underscoring the importance of world modeling in AI agents.

## 5 DISCUSSIONS

In this section, we study if the "upperbound" of our model's performance (e.g., Best-of-3 success rate) can be further improved via 1) scaling world model training data; and 2) iteratively bootstrapping "better" policy data. Finally, we quantatively measure different model's world model ability, and compare it with their overall performance.

### 5.1 SCALING WORLD MODEL TRAINING IN DYNA-THINK

Since world model learning only requires interaction data (without evaluators for task success) with the real $\mathcal{W}$, we investigate whether scaling world training in Section 3.3 can further improve policy performance. To test this, we 1) prompt GPT-4o to generate synthetic task instructions for each domain; 2) use $\pi_{\mathcal{W}}(\theta)$ collect rollout trajectories; and 3) construct a world model learning dataset following Section 3.3. Since no evaluator is required for world modeling data construction, we used all rollout trajectories that terminated within a maximum of 30 steps. This results in a total of 703 additional trajectories for world model learning. For more details please see Appendix E.3.

We present our results in Figure 4. In Figure 4b, we find that training with additional world model data steadily improves model's BoN success rate. This indicates that scaling world model training (with synthetic tasks) enhances the agent's environment understanding, enabling it to solve novel tasks. However, we did not observe a substantial increase its "robustness" - the agent did not consistently solve these novel tasks across all three trials. We believe this may be due to the stochastic nature of real-world environments, and that increasing world model training data alone cannot ensure the agent to robustly utilize all relevant world knowledge in its policy. We suggest future work should scale both world model and policy training data together, by 1) manually creating a large set of agent tasks with evaluators available for training; and/or 2) developing robust automatic evaluators (Pan et al., 2024) capable of evaluating synthetic tasks generated on-the-fly.

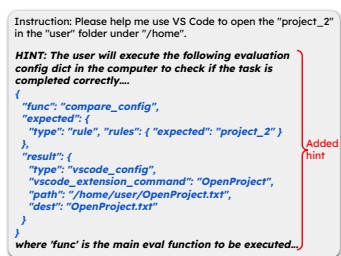

(a) Rationalization with evaluation hint.

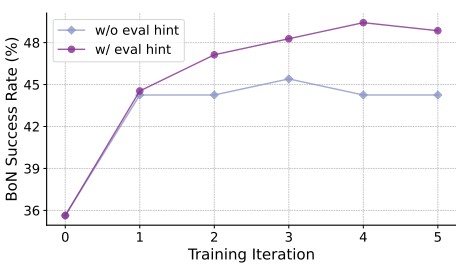

(b) Test performance.

Figure 5: Iterative policy learning with and without adding evaluation configuration as hints. The added evaluation dictionary (in blue) is part of the task configuration provided by OSWorld.

## 5.2 ITERATING POLICY TRAINING IN DYNA-THINK

We now investigate whether $\pi_{\mathcal{W}}(\theta)$ can be iteratively trained *without* any supervision from an expert LLM (e.g., GPT-4o was used to critique simulation in Section 3.3). Specifically, we follow STaR (Zelikman et al. (2022), a simple method use for math and reasoning domain) and consider two iterative training loops. **Without Evaluation Hint** (w/o eval hint), where we 1) first perform rejection sampling using $\pi_{\mathcal{W}}(\theta)$ on the training set; 2) perform policy learning on $\pi_{\mathcal{W}}(\theta)$; and 3) repeat. **With Evaluation Hint** (w/ eval hint), where for tasks that $\pi_{\mathcal{W}}(\theta)$ fail to solve during step 1, we perform "rationalization" by appending the evaluation configuration to the original instruction (Figure 5a), and then perform rejection sampling again on these tasks with evaluation hint. During training and testing, we remove the added hints from the instruction. We report test performance over 5 training iterations.

We present our results in Figure 5b. We find iterative training without rationalization (w/o eval hint) quickly plateaus; and that training with rationalization (w/ eval hint) outperforms training without rationalization. However, even when provided with evaluation configurations, we observe that $\pi_{\mathcal{W}}(\theta)$ can only solve a portion of the tasks that it fails to solve otherwise. This indicates that agent tasks, such as computer-use, remains to be a challenging domain for current language models.

## 5.3 QUANTIFYING WORLD MODEL ACCURACY

In this work, we introduced Dyna-Think to integrate world model learning with policy learning in a single $\pi_{\mathcal{W}}(\theta)$. To measure the effectiveness of this world model learning, we now evaluate the **World Model Accuracy** (Acc.) of different models, and the **Pearson Correlation Coefficient** ($r$) between the (average) world model accuracy for each task and the task success. To evaluate world model accuracy given an action $a_i = \langle \text{think}_i, \text{act}_i \rangle$ generated by $\pi_{\mathcal{W}}(\theta)$,

Table 4: Measuring the world model accuracy ($Acc$) and its correlation ($r$) with task success rate.

| Method | Policy | | World Model | |
|---|---|---|---|---|
| | Avg | BoN | Acc | $r$ |
| R1-distill-70B | 7.6 | 15.5 | 31.3 | 0.32 |
| R1 | 31.2 | 44.8 | 53.1 | 0.37 |
| DIT(R1) | 22.6 | 35.6 | 38.9 | 0.45 |
| +RFT | 23.2 | 38.5 | 46.6 | 0.37 |
| +DDT($\tilde{\mathcal{T}}_{\text{critic}}$) | 26.6 | 44.3 | 55.7 | 0.44 |

we 1) prompt GPT-4o to extract the world model simulation text from $\text{think}_i$ that corresponds to $\text{act}_i$; and 2) prompt GPT-4o judge whether the extracted simulation is correct *given the next state* $o_{i+1}$ after executing $a_i$. For each model, we calculate this for every turn in each trajectory that terminated within a maximum of 30 steps. Please refer to Appendix G.1 for prompts used and more details.

In Table 4, we find that 1) models that achieve a higher success rate also achieve a higher world model accuracy, and that 2) average world model accuracy for each task shows strong correlation with task success, with a minimum correlation of $r = 0.32$ across all models. Next, we find that DDT training significantly improved the world model accuracy (16.8% absolute) along with an improved success rate, even though it was trained on the *same set of policy data* as RFT (see Section 4.3). This shows that combining world-model and policy learning effectively boosts AI agent performance.

## 6 CONCLUSION

We present Dyna-Think, a new thinking framework that synergizes reasoning, acting, and *planning by simulating with an internal world model* to improve the performance of AI agents. Dyna-Think consists of two training stages: DIT to initialize a model with simulation ability during reasoning, and DDT for further improvement. We evaluated our methods on OSWorld and WindowsAgentArena,

and find our models based on Qwen2.5-32B-Instruct reach a similar best-of-n performance compared to R1(685B), while generating 2x less tokens on average. Our empirical analysis reveals that 1) critique-style world model training is effective for policy improvement; and 2) stronger AI agents show stronger world modeling ability. Our results suggest a promising direction for integrating world model simulation into AI agents to enhance their reasoning, planning, and acting abilities.

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

## A    LLM USAGE

This work used LLMs as general-purpose writing assistants to improve the grammar and clarity of the paper. We DO NOT use LLMs to generate research ideas, automate experiments, or analyze results.

## B    LIMITATIONS

**Training Long-Horizon Trajectories**    Computer-use tasks on benchmarks such as OSWorld (Xie et al., 2024) often require 10 and sometimes up to 100 steps to complete using current LLM (OpenAI, 2025b; Anthropic, 2025b). This makes model training challenging due to significantly increased sequence length. In this work, we train each $a_i$ by keeping a maximum context length of 20480 tokens (about 3-4 turns), and use DeepSpeed Zero-3 (Rajbhandari et al., 2020) to reduce memory usage. In general, we believe memory-efficient methods to enable long-context training will be beneficial, which we leave for future work.

**Base Model Capability**    Powerful agents based on LLMs such as DeepSeek-R1 often have model sizes reaching hundreds of billions of parameters. In this work, we used Qwen2.5-32B-Instruct as we found training smaller models (e.g., 7B in size) yields limited performance improvements. We believe this is due to the challenging nature of 1) computer-use tasks; and 2) learning long-CoT data (Li et al., 2025). To significantly improve computer-use performance for these smaller models, we believe substantial task-specific post-training may be required. We leave this for future work.

## C    ETHICS STATEMENT

Generally, while most methods and models are not designed for unethical usage, there is often potential for abuse in their applications. Computer-use agents can be used for a wide-range of tasks such as automating form filling, information gathering, software development, and more. In this work, we proposed our Dyna-Think framework to enhance the performance and (token) efficiency of AI agents. However, since computer-use agents are fundamentally task-agnostic, it is possible to use them for unethical tasks such as scamming or disseminating false information on the internet. We believe developing guardrails such as safety filters (OpenAI, 2022; Inan et al., 2023) are highly valuable for AI agent research. We do not condone the use our Dyna-Think methods for any unlawful or morally unjust purposes.

## D    DYNA-THINK IMITATION LEARNING DETAILS

We use GPT-4o to reconstruct R1-style reasoning in Section 3.2. For each turn in an R1-generated trajectory, we prompt GPT-4o to reconstruct $a_i$ given its immediate previous observation $o_{i-1}$ and relevant system instructions. We present our prompt in Table A5. A reconstructed output is shown in Figure A1. For R1 responses within 550 tokens, we do not perform abbreviation as these responses already mostly only constitutes verification, world model simulation, and action. To help ensure GPT-4o removes as much token as possible, for each prompt we sample 5 outputs, and use the shortest output in our DIT training corpus.

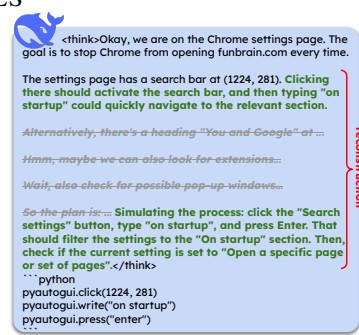

Figure A1: Example reconstruction in DIT. World simulation is in green.

In preliminary experiments, we explored alternative reconstruction methods, such as (1) directly prompting GPT-4o to remove thinking segments unnecessary for the final action without any other heuristics, and (2) focusing on other cognitive behaviors like self-reflection. We found that our emphasis on world model simulation (and initial verification) is essential for achieving strong performance while maintaining a low token count.

Table A1: Ablation on using GPT-4o

| Method | Success Rate |
|---|---|
| DIT(w/ GPT-4o) | 22.6 |
| DIT(w/ DeepSeek-V3) | 22.8 |
| DDT($\hat{\mathcal{T}}_{\text{critic}}$)(w/ GPT-4o) | 26.6 |
| DDT($\hat{\mathcal{T}}_{\text{critic}}$)(w/ DeepSeek-V3) | 27.0 |

## E    DYNA-THINK DYNA TRAINING DETAILS

### E.1    WORLD MODEL DATA PROMPTS

To construct world model data, we experimented with three methods in Section 3.3. **Next-state prediction** (DDT($\hat{\mathcal{T}}$)) which trains the model to directly predict the next state; **State-difference prediction** (DDT($\hat{\mathcal{T}}_\Delta$)) which trains the model to predict the difference $\Delta(o_i, o_{i+1})$; and **Simulation-critique generation** (DDT($\hat{\mathcal{T}}_{\text{critic}}$)), which trains the model to generate a critique for simulation in $a_i$. To obtain data for DDT($\hat{\mathcal{T}}_\Delta$) we prompt GPT-4o using prompts in Table A6. To obtain data for DDT($\hat{\mathcal{T}}_{\text{critic}}$), we first prompt GPT-4o to extract the world model simulation in $a_i$ corresponding to its final action, and the prompt it to generate a critique using prompts in Table A7. Example next-state, state-difference, and simulation-critique data is shown in Figure A3.

In this work, we primarily used GPT-4o due to its strong general capability and prevalence in many related work. In Table A1, we briefly explored using DeepSeek-V3-0324 and find that it achieves similar results, indicating that our method is not specific to the use of GPT-4o.

### E.2    TRAINING DETAILS

We present an overview of Dyna-Think Dyna Training in Figure A4. Given a set of rollout trajectories, we first perform world model training and then perform policy training. During policy training, the model $\pi_{\mathcal{W}}(\theta)$ is trained to predict the next action $a_i$ given its previous context using correct trajectories. During world model training, we experiment with three data formats: DDT($\hat{\mathcal{T}}$) directly trains the model to predict the next state $o_{i+1}$; DDT($\hat{\mathcal{T}}_\Delta$) trains the model to predict a natural language description of the state difference $\Delta(o_i, o_{i+1})$ generated by prompting GPT-4o (see Table A6; and DDT($\hat{\mathcal{T}}_{\text{critic}}$) trains the model to generate a critique for the world model simulation in $a_i$ that caused $o_{i+1}$ (see Table A7).

To better localize the critique to its corresponding world model simulation, we additionally 1) prompt GPT-4o to inject the critique back to the original response; and 2) only train on the critique by masking out all other tokens (see Figure A2). To inject a critique back into $a_i$, we 1) append an id "[[id=x]]" at the end of every sentence in $a_i$; and 2) prompt GPT-4o to output an injection location; and 3) inject the critique back into the response based on the output id.

```
<think>Okay, let's see. The user wants to find the daily forecast
for New York on AccuWeather. The current accessibility tree
shows that they are on the AccuWeather homepage. There's an
entry field labeled "Search your Address, City or Zip Code" at
position (653, 429) with size (232, 20). That's probably where I
need to type "New York" to search.
```
**Simulation**
```
First, I should click on that search bar. The coordinates for the
center of the entry would be around (653 + 232/2, 429 + 20/2) =
(653 + 116, 429 + 10) = (769, 439). So pyautogui.click(769, 439).
Then type "New York" and press enter. <judge>Wait, perhaps
after typing 'New York' into the search bar, an additional action
is needed to confirm or select the forecast for New
York.</judge>
```
**Critique**
```
After typing the city name, the forecast for New York should
load. So the next step is to click the search bar, type the city
name, and press enter.</think>
```python
pyautogui.click(769, 439)  # Click on the search bar
time.sleep(0.5)
pyautogui.write('New York')
time.sleep(0.5)
pyautogui.press('enter')
time.sleep(2)
```
```

Figure A2: Example simulation critique data. We first prompt GPT-4o to extract the world model simulation (shown in green) corresponding to the final action; and then prompt GPT-4o to generate an critique (shown in red) based on the extracted simulation and the next state (see Table A7). During training, all tokens except for text in red is masked.

---

**Algorithm 1** DDT Training (Policy + World Model with Critique Injection)

---

**Require:** Policy/world model $\pi_W(\theta)$, environment $\mathcal{T}$, rollout trajectory $\tau$, reward function $R$

1: **repeat** $N$ **times**                                                                 ▷ Data Collection:

2:      Roll out $\pi_{\mathcal{W}}(\theta)$ in $\mathcal{T}$ for trajectory $\tau = (o_0, a_1, o_1, a_2, \ldots, a_T)$

3:      Construct policy dataset $\mathcal{D}_\pi \leftarrow \{(\text{context}(o_0, a_{<i}, o_{<i}), a_i, R)\}$

4:      Construct world model dataset $\mathcal{D}_W \leftarrow \{(o_i, a_i, o_{i+1})\}$

                                                                       ▷ World Model Training:

5:      **for all** $(o_i, a_i, o_{i+1}) \in \mathcal{D}_W$ **do**

6:          **Option 1:** Next-state prediction (DDT($\hat{\mathcal{T}}$))

         Input $(o_i, a_i)$, target $o_{i+1}$

         $\mathcal{L}_{\text{wm}} \leftarrow \mathcal{L}_{\text{LM}}(\pi_W(\theta), (o_i, a_i), o_{i+1})$

7:          **Option 2:** State-change prediction (DDT($\hat{\mathcal{T}}_\Delta$))

         Input $(o_i, a_i)$, target $\Delta(o_i, o_{i+1})$ (NL description from GPT-4o)

         $\mathcal{L}_{\text{wm}} \leftarrow \mathcal{L}_{\text{LM}}(\pi_W(\theta), (o_i, a_i), \Delta(o_i, o_{i+1}))$

8:          **Option 3:** Critique prediction (DDT($\hat{\mathcal{T}}_{\text{critic}}$))

         (1) Append IDs `[[id=x]]` to each sentence in $a_i$

         (2) Use GPT-4o to output injection location

         (3) Inject critique $\rightarrow a_i'$

         (4) Mask non-critique tokens in $a_i'$

         $\mathcal{L}_{\text{wm}} \leftarrow \mathcal{L}_{\text{LM}}(\pi_W(\theta), (o_i, \ldots), \text{masked critique tokens})$

9:      **end for**

10:      Update $\theta \leftarrow \theta - \eta \nabla_\theta L_{\text{wm}}$

                                                                       ▷ Policy Training:

11:      **for all** $(\text{context}, a_i, R) \in \mathcal{D}_\pi$ **do**

12:          $\mathcal{L}_\pi \leftarrow \mathcal{L}_{\text{policy}}(\pi_W(\theta), \text{context}, a_i; R)$

13:      **end for**

14:      Update $\theta \leftarrow \theta - \eta \nabla_\theta \mathcal{L}_\pi$

15: **end**

16: **return** $\pi_{\mathcal{W}}(\theta)$

---

In Algorithm 1 we provide a high-level pseudocode for DDT. For world model training, we adopt Language Modeling(LM) Loss:

$$\mathcal{L}_{\text{LM}}(\theta) = -\sum_{t=1}^{T} \log \pi(x_t \mid x_{<t}), \tag{1}$$

During policy training, we use policy gradient to optimize the reward:

$$\nabla_\theta \mathcal{L}_{\text{policy}}(\theta) = \mathbb{E}_{\tau \sim \pi_{\mathcal{W}}(\theta)} \Big[ \sum_{t=0}^{T} \nabla_\theta \log \pi(a_t \mid o_t) R(\tau) \Big], \tag{2}$$

where we use $R(\tau) = 1$ for successful tasks, otherwise 0. In our main experiment (Table 1), we perform $N = 1$ iteration enable to have a fair comparison between the data used for policy and world model training across different algorithms (RFT and vanilla Dyna). In Section 5.2 we extend our method to multiple iterations.

### E.3 SYNTHETIC TASK GENERATION

Since world model training only requires rollout trajectories (regardless of their correctness), we experiment with scaling up world model training by using GPT-4o to generate synthetic tasks for rollouts. Specifically, we prompted GPT-4o using the template in Table A8, and generated 200 tasks for each domain. To ensure diversity between tasks, we prompt GPT-4o to directly generate $N = 10$ distinct tasks in a single prompt, and repeat this process 20 times for each domain. After this generation process, we manually inspected multiple task and did not find obvious duplicates or unreasonable tasks.

In practice, we notice that many tasks require additional configuration (e.g., setting up email profile for the Thunderbird domain, or downloading an image to edit in GIMP). To accommodate this,

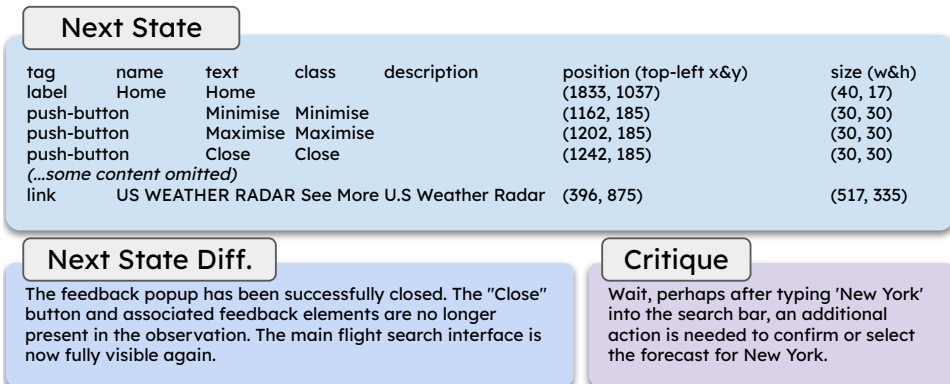

Figure A3: Example next-state, state-difference, and simulation-critique data used in DDT training.

we consider a few domain-specific changes. For all synthetic tasks in the thunderbird domain, we use the same email account configurations as used in the original OSWorld dataset. For the GIMP domain, we augmented our prompt template in Table A8 to additionally include an example GIMP task configuration from the OSWorld dataset, and instructed the model to generate new tasks based on the given configuration (see Table A9). We then use the same task configuration used in the prompt, but replace the instruction with our generated task. This process ensures that the agent has the necessary initialization for the task (e.g., at least an image to edit). For all other domains (OS, Chrome, VSCode), we use an empty configuration as they do not require additional resources.

Given these synthetic tasks, we directly perform rollouts using $\pi_{\mathcal{W}}(\theta)$. We then remove trajectories that did not terminate within a maximum of 30 steps, and use all the remaining rollouts to construct world model training data used in Section 5.1. This resulted in a total of 6467 turns available for training, 8x more than the world model training data used in Section 4.3.

## F  OSWORLD DETAILS

### F.1  ADDITIONAL DOMAINS

In Table A3, we present our model performance on domains not included in our main except (VLC, LibreOffice Impress, LibreOffice Writer, LibreOffice Calc, and Workflow). While we did observe improvement on average after training, we find these are limited (e.g., 6% improvement in VLC only corresponds to *completing 1 additional task*). Overall, we find solving tasks in these domains often encounters representation issue, such as the need to interact with a video (in VLC) or making *visual* changes to a powerpoint slide or document (in LibreOffice tasks). These visual information were often missing or mis-represented in the text-modality (accessibility tree) provided by the OSWorld benchmark. Thus, we mainly compare methods on domains such as OS, VSCode, Chrome, GIMP, and Thunderbird, where most tasks are solvable under the text-modality.

### F.2  TASK AUGMENTATIONS

As of the date of this work, OSWorld (Xie et al., 2024) is the largest-scale computer-use benchmark available with 369 tasks covering 10 domains. However, as creating evaluators for computer-use tasks is non-trivial, this benchmark mainly serves as testing performances due to its limited size for training. To this end, we follow recent work (Su et al., 2025) and increase the number of computer-use tasks. Specifically, we manually create new tasks by modifying the original instruction; and/or the task initialization configuration; and the corresponding evaluator configuration. We ensure the resulting task is 1) *distinct* from the original task, such that action sequences that can correctly complete the original task does not complete the augmented tasks; and 2) *can be correctly*

Table A2: Number of tasks per domain in the training and test set.

| Domain | Train | Test |
|--------|-------|------|
| OS | 57 | 31 |
| VSCode | 46 | 38 |
| Chrome | 88 | 54 |
| GIMP | - | 32 |
| TBird | - | 19 |

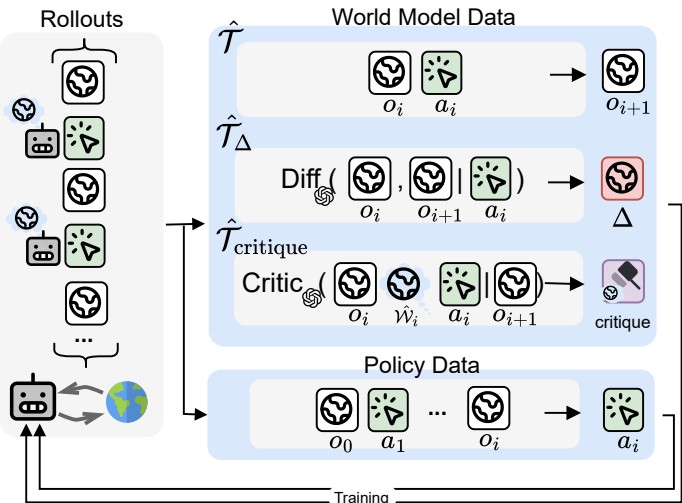

Figure A4: Illutration of Dyna-Think Dyna Training. Given a set of rollout trajectories, we first perform world model training and then perform policy training.

Table A3: Model performance on additional domains (VLC, LibreOffice Impress, LibreOffice Writer, LibreOffice Calc, and Workflow). All model training are based on Qwen-2.5-32B-Instruct. We excluded these domains from Section 4 because they often require *visual* interactions (e.g., VLC involves video controls and LibreOffice Impress often needs slide edits) that the accessibility tree often cannot represent.

| Method | Gen. Token (10%-90%) | Success Rate | | | | |
|---|---|---|---|---|---|---|
| | | VLC (17) | Impress (47) | Writer (23) | Calc (47) | Workflow (101) |
| - (Qwen2.5-32B-Instruct) | 0.2x-1.0x | 11.76 | 6.38 | 4.35 | 0.00 | 6.93 |
| DIRECT DISTILL(R1) | 2.0x-7.7x | 17.65 | 4.26 | 8.70 | 2.13 | 6.93 |
| DIT(R1) | 1.3x-2.9x | 11.76 | 2.12 | 13.04 | 2.13 | 7.92 |

*evaluated* by modifying the existing evaluator configurations. We present some example augmented tasks in Table A4.

# G  ANALYSIS DETAILS

## G.1  WORLD MODEL ACCURACY TEST

Given a trajectory from $\pi_{\mathcal{W}}(\theta)$ that performs world model simulation during its thinking process, we 1) first prompt GPT-4o to extract the world model simulation corresponding to the final action $a_i$; and then 2) prompt GPT-4o to judge the correctness of this simulation *given the next state*. We repeat this for every $a_i$ in a all trajectories that terminated within a maximum budget of 30 steps. We present the prompts used to extract world model simulation in Table A10, and prompts used to measure simulation accuracy in Table A11.

Table A4: Example augmented tasks. We modify the task instruction and/or the initialization configuration, such that action sequences that can correctly complete the original task does *not* complete the augmented tasks.

| Domain | Task |
|---|---|
| OS | // original task
Can you remove the first favorite app from 'favorites'?" |
| | // augmented tasks
Can you remove thunderbird and google chrome from the 'favorites' apps?
Can you remove all the favorite apps except for thunderbird from 'favorites'?" |
| VSCode | // original task
Please help me install the autoDocstring extension in VS Code." |
| | // augmented tasks
Please help me install the Live Server extension by Ritwick Dey in VS Code.
Please help me install the Auto Docstring and the Docker extension in VS Code. |
| Chrome | // original task
Find a men's T-Shirt that is in large size with a stripe pattern, short sleeve and under the Sales&Discount. |
| | // augmented tasks
Find a men's T-Shirt that is blue, sleeveless and in XL size on Macy's website.
Find a listing of woman's sandals with ankle strap on Macy's website." |

Table A5: Prompts used abbreviate thinking in DIT. Generated response is shown in blue.

| Role | Prompt |
|---|---|
| System | You are an agent which follow my instruction and perform desktop computer tasks as instructed (...some content omitted)
You are asked to complete the following task: (...some content omitted) |
| User | Given the info from accessibility tree as below:

tag name text class description position (top-left x&y) size (w&h)
label Home Home (1833, 1037) (40, 17)
(...some content omitted)
push-button Trash "" (0, 784) (70, 64)
toggle-button Show Applications "" (0, 1010) (70, 70)

What's the next step that you will do to help with the task? |
| Assistant | <think>Okay, the user is trying to (...some content omitted)</think>
"'python
pyautogui.click(x=35, y=1045) # Click Show Applications
time.sleep(1)
pyautogui.write('terminal')
(...some content omitted)
"' |
| User | ACTIONS TO BE EXECUTED:
"'python
pyautogui.click(x=35, y=1045) # Click Show Applications
time.sleep(1)
pyautogui.write('terminal')
(...some content omitted)
"'

Your task now is to ABBREVIATE the THINKING process (inside the <think> xxx </think> part) of the previous response.
Specifically, you need to KEEP THE FOLLOWING INFORMATION in the thinking section, while removing others as much as possible:
1. Thoughts related to verification: checking whether previous action has succeeded/failed. This is usually the first paragraph in the THINKING process.
2. Thoughts related to what could be potential actions to do next, if mentioned. This usually follows after the verification process, (...some content omitted)
3. Thoughts related to final action simulation: simulating the outcome/process of the 'ACTIONS TO BE EXECUTED' section (...some content omitted)
4. Other thoughts that you believe is necessary to logically connect step 2 to step 3 above.
5. Implementation details such as code snippets related to part 2 and 3 above should also be KEPT.

{{one shot example}}

Now, abbreviate the thinking process in the <think> xxx </think> section of the previous response. Your output should:
- keep the **ORIGINAL FORMATTING AND TONE** of the response (...some content omitted)
- only **REMOVE UNNECESSARY THINKING PARTS WIHOUT TOUCHING ANYTHING ELSE**. You can ONLY rephrase a few sentences if they are necessary (...some content omitted)
- logically **CONNECT to the 'ACTIONS TO BE EXECUTED' section**(...some content omitted)
Respond in the following format. DO NOT generate anything else. Abbreviated thinking process: <think>Okay, ...(your abbreviated thinking process)...</think> |
| Assistant | <think>Okay, the user is trying to (... some content omitted)</think> |

Table A6: Prompts used to obtain $\Delta(o_i, o_{i+1})$ data. Generated response is highlighted in blue.

| Role | Prompt |
| --- | --- |
| System | You are an agent which follow my instruction and perform desktop computer tasks as instructed (...some content omitted)
You are asked to complete the following task: (...some content omitted) |
| User | Given the info from accessibility tree as below:

tag name text class description position (top-left x&y) size (w&h)
label Home Home (1833, 1037) (40, 17)
(...some content omitted)
toggle-button Show Applications "" (0, 1010) (70, 70)

What's the next step that you will do to help with the task? |
| Assistant | `<think>`Okay, the user is trying to (...some content omitted)`</think>`
```python
pyautogui.click(x=35, y=1045) # Click Show Applications
time.sleep(1)
pyautogui.write('terminal')
(...some content omitted)
``` |
| User | ACTIONS TO BE EXECUTED:
```python
pyautogui.click(x=35, y=1045) # Click Show Applications
time.sleep(1)
pyautogui.write('terminal')
(...some content omitted)
```

NEXT OBSERVATION AFTER EXECUTION:
tag name text class description position (top-left x&y) size (w&h)
label Home Home (1833, 1037) (40, 17)
(...some content omitted)
toggle-button Show Applications "" (0, 1010) (70, 70)

Your task now is to describe changes in the next observation caused by the previous action. Specifically, you need to describe what changes, if any, occured on the 'NEXT OBSERVATION AFTER EXECUTION' after executing the 'ACTIONS TO BE EXECUTED'. The description should:
1. give high-level and functional description in detail after the action.
2. focus solely on describing the changes that are caused by the action.
3. if the 'NEXT OBSERVATION AFTER EXECUTION' is identical to the previous observation, say 'no changes'.
4. keep your description concise. **DO NOT explain why** certain changes ocurred, only describe the changes in observation.

Now, describe the changes in 'NEXT OBSERVATION AFTER EXECUTION' that are likely caused by 'ACTIONS TO BE EXECUTED' section. Respond in the following format. DO NOT generate anything else. Keep your answer within 3 sentences.
Changes description: \<your description of the changes\> |
| Assistant | Changes description: The terminal has been (...some content omitted) |

Table A7: Prompts used to obtain Critic$(o_i, a_i | o_{i+1})$ data. Generated response is highlighted in blue. Extracted world model simulation from the agent's response is shown in green.

| Role | Prompt |
|------|--------|
| System | You are an agent which follow my instruction and perform desktop computer tasks as instructed (...some content omitted) 
 You are asked to complete the following task: (...some content omitted) |
| User | Given the info from accessibility tree as below: 

 tag name text class description position (top-left x&y) size (w&h) 
 label Home Home (1833, 1037) (40, 17) 
 (...some content omitted) 
 toggle-button Show Applications "" (0, 1010) (70, 70) 

 What's the next step that you will do to help with the task? |
| Assistant | \<think\>Okay, the user is trying to (...some content omitted)\</think\> 
 "'python 
 pyautogui.click(x=35, y=1045) # Click Show Applications 
 time.sleep(1) 
 pyautogui.write('terminal') 
 (...some content omitted) 
 "' |
| User | ACTIONS TO BE EXECUTED: 
 "'python 
 pyautogui.click(x=35, y=1045) # Click Show Applications 
 time.sleep(1) 
 pyautogui.write('terminal') 
 (...some content omitted) 
 "' 

 RELATED THINKING FROM PREVIOUS RESPONSE: 
 Clicking on the 'Show Applications' will open an application grid. Then, we can (...some content omitted). 

 NEXT OBSERVATION AFTER EXECUTION: 
 tag name text class description position (top-left x&y) size (w&h) 
 label Home Home (1833, 1037) (40, 17) 
 (...some content omitted) 
 toggle-button Show Applications "" (0, 1010) (70, 70) 

 Your task now is to JUDGE if the action (in the 'ACTIONS TO BE EXECUTED' section) achieved the intended effect in the 'ACTUAL NEXT STATE' section. Specifically, you need to: 
 1. decide whether the 'RELATED THINKING FROM PREVIOUS RESPONSE' section is correct based on the 'ACTUAL NEXT STATE' section. 
 2. provide a confidence score from 0 to 5 for the above judgment, where 0 means 'not at all confident' and 5 means 'very confident'. 
 3. if you think 'RELATED THINKING FROM PREVIOUS RESPONSE' is incorrect, provide a one-sentence reason (...some content omitted) 
 4. if you thin the 'RELATED THINKING FROM PREVIOUS RESPONSE' is correct, provide a one-sentence reason (...some content omitted) 
 NOTE: if the thinking is CORRECT, then the hint should: 
 - (...some content omitted) - start with 'Yes, ...'. 
 NOTE: if the thinking is INCORRECT, then the hint should: 
 - (...some content omitted) - start with 'Wait, ...'. (...some content omitted) |
| Assistant | is_simulation_correct: False 
 confidence: 5 
 reason: Terminal application is not present in the next state (...some content omitted) 
 in_context_hint: Wait, maybe we need to wait a bit before typing 'terminal', or ... |

Table A8: Prompts used to generate synthetic task instructions in domains excluding GIMP. Generated response is highlighted in blue, with {*domain*} set to chrome.

| Role | Prompt |
| --- | --- |
| User | You are a helpful assistant. Imagine you are in an Ubuntu VMware environment, with access to applications like os, vscode, chrome, gimp, thunderbird, etc. Your mission is to generate a list of **10** task instructions that can be performed in this environment, and the tasks should be related to a given domain (application), which means the tasks can be solved mainly by this application. You should guarantee: (1) The tasks should be clear to understand and should be related to common activities that a user might perform in a virtual machine environment. (2) The tasks should not be exploratory, i.e. whether the tasks are completed correctly should be verifiable. You should avoid using words like "set a secure password", "install a specific software", etc. Instead, you should explicitly nominate these by making up them or using your knowledge. (3) The tasks should be interesting and challenging, but not too complicated, i.e. a user may accomplish them in a few minutes. (4) You don't need to consider the exact environment, such as the file structure and the application versions, etc. You can add details to the tasks as long as they look reasonable and not too complicated. (5) The list of tasks should be diverse and cover a range of different activities related to the given domain. (6) You should NOT add indexes or titles to the tasks. Directly list the task instructions split by new lines. The domain for the tasks is ""{*domain*}"". |
| Assistant | Access the Chrome settings and enable "Do Not Track" requests for all websites. |
| | Open Chrome and go to the YouTube website. Search for a video on sports topic and add it to the "Watch Later" playlist to ensure you can access it later. |
| | Set up Chrome to automatically translate web pages written in a foreign language to English using the built-in translation feature. |
| | Use Chrome to download a PDF version of a popular Ubuntu guide, ensuring it's saved to the "Downloads" folder. |
| | Use the "Save as PDF" feature in Chrome to save an online recipe from a cooking website to your desktop. |
| | Enable the reading list feature in Chrome's settings. Add three different articles from any technology news website to the reading list and mark one as read. |
| | Open Chrome and navigate to a tech news website like TechCrunch. Bookmark the homepage and create a new folder named "Tech News" in your bookmarks bar for future reference. |
| | Open Chrome and navigate to a weather website like weather.com. Search for the weather forecast of your city for the next week and take a screenshot of the results. |
| | Access Chrome's "Task Manager" by clicking "More tools" and "Task Manager." Identify the process consuming the most memory and end the task to free up resources. |
| | Open Chrome and navigate to the Chrome Web Store. Install the Grammarly extension and verify that it is enabled by checking the extensions list in the browser settings. |

Table A9: Prompt used to generate synthetic tasks for the GIMP domain.

| GIMP Prompt |
| --- |
| You are a helpful assistant. Imagine you are in an Ubuntu VMware environment, with access to applications like os, vscode, chrome, gimp, thunderbird, etc. |
| The followings are an example of a pair of task instruction and corresponding environment setup configuration for GIMP: |
| [Environment setup configuration]: {{config}} |
| [Task instruction]: {{instruction}} |
| Your mission is to generate a list of 10 task instructions which are also related to gimp and can all be performed with the SAME environment setup configuration as the above example. |
| You should guarantee: The tasks should be clear to understand and should be related to realistic activities. The tasks could be interesting, challenging, and properly complicated, i.e. a user may accomplish them in a few minutes. |
| If the example's configuration is empty, you should ONLY consider tasks that does not require any image operations, such as starting from a blank canvas, or changing the settings of gimp itself. |
| For the result you generate, please make sure: |
| The list of tasks should be diverse and cover a range of different activities related to gimp. They should be DISTINCT from each other and the examples provided. |
| You should NOT add indexes or titles to your answers. Directly list the task instructions split by new lines. |

Table A10: Prompts used to extract world model simulation. Generated response is in blue.

| Role | Prompt |
|---|---|
| System | You are an agent which follow my instruction and perform desktop computer tasks as instructed (...some content omitted) 
 You are asked to complete the following task: (...some content omitted) |
| User | Given the info from accessibility tree as below: 

 tag name text class description position (top-left x&y) size (w&h) 
 label Home Home (1833, 1037) (40, 17) 
 (...some content omitted) 
 toggle-button Show Applications "" (0, 1010) (70, 70) 

 What's the next step that you will do to help with the task? |
| Assistant | \<think\>Okay, the user is trying to (...some content omitted)\</think\> 
 ```python 
 pyautogui.click(x=35, y=1045) # Click Show Applications 
 time.sleep(1) 
 pyautogui.write('terminal') 
 (...some content omitted) 
 ``` |
| User | ACTIONS TO BE EXECUTED: 
 ```python 
 pyautogui.click(x=35, y=1045) # Click Show Applications 
 time.sleep(1) 
 pyautogui.write('terminal') 
 (...some content omitted) 
 ``` 


 Your task now is to ANNOTATE the thinking process of the previous action. Specifically, you need to EXTRACT the simulation process behind the 'ACTIONS TO BE EXECUTED' section above. You should find them inside the previous response (e.g., inside \<think\> xxx \</think\>). 
 The extracted simulation process: 
 1. SHOULD mention the high level plan of these actions, if exists 
 2. SHOULD mention the ALL the action in the 'ACTIONS TO BE EXECUTED' section AND the effect each action was supposed to have. If this is missing for some actions, you can fill in your best guess based on the context. 
 3. SHOULD ONLY mention any action or effect included in the 'ACTIONS TO BE EXECUTED' section. 
 4. SHOULD keep the original formatting and tone of the response. 
 5. If you DID NOT FIND ANY such simluation process exists in the previous response, you should still follow rule 2 but set the 'has_world_model_simulation' to False. 

 FOR EXAMPLE, given the following action and original response: 
 {{one shot example}} 

 Now, extract the simulation process for the 'ACTIONS TO BE EXECUTED' section. |
| Assistant | has_world_model_simulation: True 
 simulation_process: Clicking 'Show Applications' should open an application grid, with search bar at... In the search bar we can type (...some content omitted) |

Table A11: Prompts used to test simulation accuracy. World model simulation is in green. Generated response is in blue.

| Role | Prompt |
|---|---|
| System | You are an agent which follow my instruction and perform desktop computer tasks as instructed (...some content omitted)

You are asked to complete the following task: (...some content omitted) |
| User | Given the info from accessibility tree as below:

tag name text class description position (top-left x&y) size (w&h)
label Home Home (1833, 1037) (40, 17)
(...some content omitted)
toggle-button Show Applications "" (0, 1010) (70, 70)

What's the next step that you will do to help with the task? |
| Assistant | <think>Okay, the user is trying to (...some content omitted)</think>
```python
pyautogui.click(x=35, y=1045) # Click Show Applications
time.sleep(1)
pyautogui.write('terminal')
(...some content omitted)
``` |
| User | RELATED THINKING FROM PREVIOUS RESPONSE:
Clicking on the 'Show Applications' will open an application grid. Then, we can (...some content omitted).

ACTIONS TO BE EXECUTED:
```python
pyautogui.click(x=35, y=1045) # Click Show Applications
time.sleep(1)
pyautogui.write('terminal')
(...some content omitted)
```

ACTUAL NEXT STATE:
tag name text class description position (top-left x&y) size (w&h)
label Home Home (1833, 1037) (40, 17)
(...some content omitted)
toggle-button Show Applications "" (0, 1010) (70, 70)

Your task now is to JUDGE if the action (in the 'ACTIONS TO BE EXECUTED' section) achieved the intended effect in the 'ACTUAL NEXT STATE' section. Specifically, you need to:
1. decide whether the 'RELATED THINKING FROM PREVIOUS RESPONSE' section is correct based on the 'ACTUAL NEXT STATE' section.
2. provide a confidence score from 0 to 5 for the above judgment, where 0 means 'not at all confident' and 5 means 'very confident'.
3. if you think the 'RELATED THINKING FROM PREVIOUS RESPONSE' is incorrect, provide a one-sentence reason of which part of the thinking may cause the error.
4. if you think the 'RELATED THINKING FROM PREVIOUS RESPONSE' is correct, provide a one-sentence reason why you think so. |
| Assistant | is_simulation_correct: False
confidence: 5
reason: Terminal application is not present in the next state (...some content omitted) |

