# OpenReview forum: "Dyna-Think: Synergizing Reasoning, Acting, and World Model Simulation in AI Agents"
_ICLR.cc/2026/Conference — Submitted to ICLR 2026_

### Official Review · Reviewer_yrjn · 2025-10-29

**Soundness:** 3
**Presentation:** 3
**Contribution:** 3
**Rating:** 6
**Confidence:** 3

**Summary:**

This paper proposes the Dyna-Think framework, which integrates the world model with the reasoning process of agents to enhance the decision-making performance of long-term computer-use task. Firstly, by reconstructing the reasoning process of DeepSeek-R1, Dyna-Think imitates the reasoning and world modeling related to the final action. Then, Dyna-Think trains the policy and world model using the world model dataset and the policy dataset, adopting critique-style training for the world model. The experimental results show that critique-style world model training is effective; world simulation is crucial; and the world model's ability is strongly correlated with the success rate of tasks. Contributions: Dyna-Think with 32B model achieves the performance of the 685B R1, with a 50% reduction in tokens, proving the synergistic effect of world model learning and policy learning.

**Strengths:**

1. The motivation is clear. Starting from the observation that existing reasoning models tend to overthink. The authors identify that world model simulation is the most influential part for the final action, leading to a framework that integrates reasoning, acting, and world modelling.

2. Dyna-Think extends traditional Dyna algorithms, which separate policy π(θ) and world model W(μ), into a unified model πW(θ) that internalizes world model simulation into the agent's thinking process, which is innovative.

3. The experiment is detailed, validating the effectiveness of Dyna-Think and exploring different world model training objectives, different thinking behaviors, the world model data scale and the correlation between world model accuracy and agent performance.

4. The paper is well-written with clear motivation, method description, and experimental results. The figures effectively illustrate key concepts (e.g., Figure 1 comparing R1 and Dyna-Think, Figure 2 showing the framework), and the experimental results are presented systematically with appropriate ablations and analysis.

**Weaknesses:**

Refer to questions section.

**Questions:**

1. Dyna-Think adopts Qwen2.5-32B-Instruct as the base model and DeepSeek-R1 as the teacher model. Is the current model's performance ceiling primarily constrained by R1's performance limit, or is it limited by the scale of the base model? Could training a larger base model further improve performance and potentially surpass R1?

2. Dyna-Think uses GPT-4o to reconstruct the thinking process (DIT) and generate critiques for world model training (DDT). Could the use of a single model lead to biased datasets and training processes? Would adopting different LLMs of varying scales or with different reasoning styles affect Dyna-Think's results?

3. Critique-based world model training may lead to overfitting to the training environment. As shown in Lines 338 and 340, critique-based world model training improves in-domain (ID) performance but shows a drop in out-of-domain (OOD) performance compared to next-state prediction. Additionally, as shown in Lines 462 and 465, Dyna-Think achieves higher world model accuracy but lower task success rate in certain cases. What causes this discrepancy? Does this suggest that critique-based world model training leads to overfitting to the training environment? Further analysis on the generalization properties of different world model training objectives would be valuable.

4. The experimental results shown in Section 5.1 (Figure 4b) appear to report performance on the full test set (All: 174 tasks). Could the authors provide a breakdown showing the impact of synthetic data scaling separately on ID and OOD performance? This would help understand whether scaling world model training with synthetic tasks primarily benefits ID generalization, OOD generalization, or both, and provide insights into the data efficiency of world model learning.

---

> ### Author Response · Authors · 2025-11-17
> **Response to Weaknesses and Questions**
>
> We thank reviewer yrjn for acknowledging our clear motivation, “innovative” method, detailed experiments, and “well-written” method descriptions and results!
>
> ---
>
> > Dyna-Think adopts Qwen2.5-32B-Instruct as the base model and DeepSeek-R1 as the teacher model. Is the current model's performance ceiling primarily constrained by R1's performance limit, or is it limited by the scale of the base model? Could training a larger base model further improve performance and potentially surpass R1?
>
> We agree that base model capability is crucial, as overall performance is generally tied to it. However, unfortunately we do not have the required compute to train a larger model than Qwen2.5-32B-Instruct (e.g., Qwen2.5-72B-Instruct).
>
> Regarding the teacher model, we chose DeepSeek-R1 because, at the time of our work, it was one of the strongest reasoning models that exhibited world modeling ability. This was a critical factor; other models were unsuitable (e.g., GPT-4o is not a reasoning model, and GPT-o1/o3 does not provide its reasoning traces in its API outputs). While we believe a stronger teacher could curate better offline training data for DIT, we leave this exploration for future work.
>
> \
>
> > Dyna-Think uses GPT-4o to reconstruct the thinking process (DIT) and generate critiques for world model training (DDT). Could the use of a single model lead to biased datasets and training processes?
>
> Thanks for this question. We performed both DIT and DDT(critic) training using DeepSeek-V3 instead of GPT-4o below. In general, we find these results are similar to that presented in Table 1-2. This suggests our framework's benefits are not specifically dependent on GPT-4o.
>
>
> | Method | OSWorld Success Rate |
> | :--- | :--- |
> | DIT(w/ GPT-4o) | 22.6 |
> | DIT(w/ DeepSeek-v3) | 22.8 |
> | DDT(critic w/ GPT-4o) | 26.6 |
> | DDT(critic w/ DeepSeek-v3) | 27.0 |
>
>
> We added this discussion and table in our revised manuscript.
>
> \
>
> > Critique-based world model training may lead to overfitting to the training environment. As shown in Lines 338 and 340, critique-based world model training improves in-domain (ID) performance but shows a drop in out-of-domain (OOD) performance compared to next-state prediction. Additionally, as shown in Lines 462 and 465, Dyna-Think achieves higher world model accuracy but lower task success rate in certain cases. What causes this discrepancy? Does this suggest that critique-based world model training leads to overfitting to the training environment?
>
>
> In OOD settings (both the OSWorld OOD test set and WindowsAgentArena), we did observe that DDT(T) outperforms DDT(critic) in Tables 1-2. We hypothesize this is because OOD environments place a greater emphasis on correctly understanding the current state, which is needed for effective planning. The next-state prediction objective in DDT(T) is thus more effective for this scenario, especially when the model's base capability is not strong. We will emphasize this more on our L358-361.
>
> In general, we note that performance drop in OOD is common even for models such as the untrained R1 (OOD score dropped 43% relative to its ID performance). Our OOD test split is constructed based on different, more challenging domains (GIMP and Thunderbird) in OSWorld, which we believe is the reason for the performance drop.
>
> \
>
> > The experimental results shown in Section 5.1 (Figure 4b) appear to report performance on the full test set (All: 174 tasks). Could the authors provide a breakdown showing the impact of synthetic data scaling separately on ID and OOD performance?
>
> Yes. We provide the results in tabular form below. In general, we do see that OOD performance plateaus faster than ID. We believe this is reasonable, since our OOD settings covered more challenging domains compared to the ID test set (see previous response).
>
> | \|w\| |  All  | ID  | OOD |
> | :--- | :--- | :--- | :--- |
> | 1x |  44.3  |  49.6 |  31.4 |
> | 3x |  46.0  |  47.2 |  43.1 |
> | 5x |  47.1  |  49.6 |  41.2 |
> | 8x |  49.4  |  52.0 |  43.1 |

---

### Official Review · Reviewer_dbiP · 2025-10-30

**Soundness:** 2
**Presentation:** 2
**Contribution:** 2
**Rating:** 4
**Confidence:** 4

**Summary:**

The paper proposes Dyna-Think, a framework that aims to integrate world model simulation into an LLM agent's reasoning process. The method consists of two main parts: Dyna-Think Imitation Learning (DIT), where the reasoning process of a large expert model (DeepSeek-R1) is "reconstructed" and abbreviated using GPT-4o to distill a more efficient thinking style into a smaller model; and Dyna-Think Dyna Training (DDT), an online fine-tuning stage that uses environment rollouts to train the agent on both policy and world model objectives.

**Strengths:**

The work tackles the critical trade-off in AI agent design by proposing a framework to distill the capabilities of large expert models into smaller, task-optimised agents. This approach positions itself as a middle ground, mediating between the high computational cost of test-time search (e.g., MCTS) and the high token cost of the elaborate reasoning found in leading thinking models.

The authors have employed a solid evaluation setup by using an held-out in-domain test sets, completely out-of-domain application suites (GIMP, Thunderbird), and a different out-of-domain operating system (WindowsAgentArena). The empirical results are indeed strong and provide a compelling case for the proposed methodology.

The empirical support for the proposed framework is robust, featuring a comprehensive evaluation that is not limited to aggregate performance metrics but also includes valuable ablations and a quantitative analysis of the agent's underlying reasoning.

**Weaknesses:**

The claim of achieving performance similar to the R1 expert warrants a more nuanced analysis, as the aggregate metrics conceal significant weaknesses in generalization and reliability. For instance, the model's reliability drops precipitously in unfamiliar contexts: its Average Success Rate falls from 30.3% on in-domain tasks to just 17.6% out-of-domain. This suggests that the agent's learned 'world model' is highly specialized to the training applications and does not transfer effectively. This issue of brittleness is also reflected in the large gap between the agent's high Best-of-N scores and its much lower Average Success rates across all domains. A comprehensive analysis would need to address these points directly, as they are central to understanding the practical limitations and the true generalization power of the proposed framework.

While the "Dyna" framing is evocative, it is also confusing. Classic Dyna involves learning a separate world model to generate simulated experiences for planning, which then augments model-free learning. The proposed method does neither of these. A more accurate framing would be "Distilling Planning into Reasoning." The DIT stage distils R1's verbose reasoning into a more structured, plan-like format, and the DDT stage uses online data to further refine this internalised planning capability. The current framing obscures the core contribution, which is about the structure of reasoning, not a classic learning-planning loop.

The results show that DDT critic performs best on the in-domain OSWorld benchmark, while next-state prediction DDT T hat performs best on the OOD WindowsAgentArena benchmark. The authors briefly hypothesize that next-state prediction might enhance the model's fundamental understanding of the environment, which is useful for novel states. This is a plausible but crucial point that deserves a much deeper discussion. It complicates the paper's narrative and suggests that the optimal form of world model training may not be universal, which is a significant practical limitation.

A key methodological concern is the framework's deep dependency on the proprietary GPT-4o model, which serves multiple roles as an editor, critic, and evaluator. This makes it difficult to attribute performance gains solely to the proposed Dyna-Think architecture versus the distilled intelligence of the powerful GPT-4o oracle. The paper would be significantly strengthened by an ablation study that investigates this dependency. For instance, demonstrating the framework's effectiveness when using a different model to generate critiques would help isolate the contribution of the architectural choices and assess the method's broader applicability.

**Questions:**

The results show a significant gap between the proposed model's average success rate and that of the R1 expert, particularly the collapse in reliability from 30.3% in-domain to 17.6% out-of-domain. Could you provide a more detailed analysis of what contributes to this performance gap and brittleness? Does this suggest the framework is more effective at domain-specific optimization than at learning generalizable skills?

Regarding the methodology, could you discuss the rationale for using GPT-4o as the central oracle over the in-study expert, R1? To what extent are the framework's successes tied to the specific SOTA capabilities of GPT-4o, and would the results be replicable with a different critic model?

The results indicate that different DDT objectives are optimal for in-domain versus out-of-domain tasks. Could you expand on your hypothesis for why this occurs, and does this imply that a robust generalist agent might require a curriculum of world model objectives rather than a single approach?

To better understand the model's behavior, could you please provide the scatter plot that visualizes the correlation between per-task World Model Accuracy and Task Success from which the Pearson correlation in Table 4 was calculated? A more granular breakdown of performance by task category would also be helpful to identify systematic strengths and weaknesses.

The "Dyna" framing is potentially confusing given the departure from the classic architecture. Have you considered framing the contribution more directly as a method for "Distilling Planning into Reasoning," which seems to more accurately capture the DIT and DDT processes?

---

> ### Author Response · Authors · 2025-11-17
> **Response to Weaknesses**
>
> We thank reviewer dbiP for acknowledging our “solid evaluation setup” including both in-domain and out-of-domain results, and “robust and comprehensive” evaluation and analysis!
>
>
> ---
>
>
> > The claim of achieving performance similar to the R1 expert warrants a more nuanced analysis, as the aggregate metrics conceal significant weaknesses in generalization and reliability. For instance, the model's reliability drops precipitously in unfamiliar contexts: its Average Success Rate falls from 30.3% on in-domain tasks to just 17.6% out-of-domain. This suggests that the agent's learned 'world model' is highly specialized to the training applications and does not transfer effectively.
>
>
> We disagree that these results indicate our learned world model is specialized to the training settings. As shown in Table 1, this performance drop is not unique to our model. For instance, the untrained R1 baseline exhibited a near-identical drop in the OOD setting (43% relative to its ID performance). Our OOD test split is constructed based on different, more challenging domains (GIMP and Thunderbird) in OSWorld, which we believe is the reason for the performance drop.
>
> \
>
> > While the "Dyna" framing is evocative, it is also confusing. Classic Dyna involves learning a separate world model to generate simulated experiences for planning, which then augments model-free learning… A more accurate framing would be "Distilling Planning into Reasoning."
>
> Motivated by Dyna, Dyna-think uses world simulation to improve reasoning and planning. Unlike Dyna (a baseline we compared to in Table 2) which trains a separate world model to improve policy, our approach *integrates world model simulation into LLM’s reasoning* and performed both policy and world model training within a single model. In Table 2, we showed that our DDT(critic) training outperforms vanilla Dyna by 2.5% and 8.7% absolute on Average and BoN success rate, respectively.
>
> In general, we believe our Dyna-think method presents a new way to view “Dyna” methods in the era of large reasoning models. However, we are open to discussion if this name is confusing.
>
> \
> > The results show that DDT critic performs best on the in-domain OSWorld benchmark, while next-state prediction DDT T hat performs best on the OOD WindowsAgentArena benchmark. The authors briefly hypothesize that next-state prediction might enhance the model's fundamental understanding of the environment, which is useful for novel states. This is a plausible but crucial point that deserves a much deeper discussion.
>
>
> Critiquing is a two-step process: the model must first generate a good initial simulation of the next state and then refine it (if incorrect). In contrast, next-state prediction only requires the model to generate a good simulation.
>
> The former is thus more difficult and may require more model capacity to learn effectively. However, once learned, its performance is strong, as shown by our ID results. In general, we believe that when training a model with *limited* capacity, the simpler next-state prediction task is more effective than critique training.
>
> \
> > A key methodological concern is the framework's deep dependency on the proprietary GPT-4o model, which serves multiple roles as an editor, critic, and evaluator.
>
> Could you clarify what you mean by 1) “deep dependency on the proprietary GPT-4o”, and 2) GPT-4o serving as "evaluator"?
>
> For 1), did you mean that our framework should try models other than GPT-4o? If so, we tried DIT and DDT(critic) with DeepSeek-V3 below. We find these results are consistent, showing a similar performance to that presented in Table 1-2. This suggests our framework's benefits are not dependent on a specific model. We have added this discussion in our revised manuscript.
>
> | Method | OSWorld Success Rate |
> | :--- | :--- |
> | DIT(w/ GPT-4o) | 22.6 |
> | DIT(w/ DeepSeek-v3) | 22.8 |
> | DDT(critic w/ GPT-4o) | 26.6 |
> | DDT(critic w/ DeepSeek-v3) | 27.0 |
>
>
> For 2), we note that we did not use GPT-4o as evaluator/reward function in any part of our DIT/DDT training. Could you please elaborate?

---

> ### Author Response · Authors · 2025-11-17
> **Response to Questions**
>
> > The results show a significant gap between the proposed model's average success rate and that of the R1 expert, particularly the collapse in reliability from 30.3% in-domain to 17.6% out-of-domain. Could you provide a more detailed analysis of what contributes to this performance gap and brittleness?
>
> Please refer to our response to “The claim of achieving performance similar to the R1 expert warrants a more nuanced analysis…”.
>
>
> \
>
> > Regarding the methodology, could you discuss the rationale for using GPT-4o as the central oracle over the in-study expert, R1? To what extent are the framework's successes tied to the specific SOTA capabilities of GPT-4o, and would the results be replicable with a different critic model?
>
> We used GPT-4o for most data processing due to its strong general capability and its widespread use in prior work [1-3]. Regarding “framework’s success tied to GPT-4o”, please refer to our response to “A key methodological concern is the framework's deep dependency on the proprietary GPT-4o…” where we showed using DeepSeek-V3 also achieves similar results.
>
> —
>
> References
>
> [1] Xu, Yiheng, et al. "Aguvis: Unified pure vision agents for autonomous gui interaction." arXiv preprint arXiv:2412.04454 (2024).
>
> [2] Xie, Tianbao, et al. "Scaling Computer-Use Grounding via User Interface Decomposition and Synthesis." arXiv preprint arXiv:2505.13227 (2025).
>
> [3] Yang, Yuhao, et al. "Aria-ui: Visual grounding for gui instructions." Findings of the Association for Computational Linguistics: ACL 2025. 2025.

---

> ### Author Response · Authors · 2025-11-24
>
> Dear reviewer dbiP,
>
> Thank you again for these questions and feedback! Please let us know at your earliest convenience if you have any further questions or concerns, or would like to us conduct any additional experiments.

---

### Official Review · Reviewer_G1k1 · 2025-10-31

**Soundness:** 2
**Presentation:** 2
**Contribution:** 2
**Rating:** 4
**Confidence:** 4

**Summary:**

This paper introduces Dyna-Think, a way to get language agents to perform world modeling (1) through imitation learning on reconstructed trajectories (Dyna-Think Imitation Learning) and (2) policy learning with auxiliary world modeling objectives (Dyna-Think Dyna Training). The authors evaluate on OSWorld and WindowsAgentArena and generally find their proposed approach to outperform the training-based baselines. In addition, the authors provide insights on how to synthetically scale up world model learning and evaluate world modeling accuracy.

**Strengths:**

1. The paper aims to tackle a timely topic as both language agents and world models have shown promise in their own respective domains, but they have not been super successfully combined so far.
2. I like that the paper measures both in-domain and out-of-domain performance.
3. The results tend to suggest some of the proposed modifications indeed help performance over the trained baselines.
4. Figure 4 seems like a nice result since it implies you can continue learning mostly (or purely?) through world model training.

**Weaknesses:**

1. Section 3.2: Based on Table A4 in the appendix, I’m not convinced that the benefits seen from doing imitation learning on R1 thinking reconstructions have anything to do with world modeling simulation. Specifically, did the authors try simpler prompts than the one listed in Table A4, maybe just keeping “remove unnecessary thinking parts without touching anything else”. In other words, how much of the benefit is just coming from generally letting GPT-4o clean up the thinking trace?
2. Figure 2 could be improved in terms of clarity: what’s the difference between Dyna Training (b) and Dyna-Think Training (c). Also, it seems (a) is just showing a general RL diagram - it’s unclear to me how this helps explain the proposed method?
3. Page 5: “then constructing a policy learning dataset that trains π_W(θ) to predict each action a_i given the previous context context (o0,a<i,o<i) based on a reward function (e.g., task success).” → Could the authors be more specific here? Predicting each action given the previous context sounds like behavioral cloning, but the authors also mention a reward so I’m not sure exactly what objective is being optimized?
4. The paper could also use some improvements in writing. There are several sentences that contain spelling or grammar errors.
5. Section 3.3: why does providing a critique generated by GPT-4o have anything to do with world modeling?
6. Section 3.3: a lot of the algorithm is described in words, and it’d greatly help the clarity of exposition if some of this was written out more formally.
7. Page 7: “… and that focusing on/improving simulation ability is effective at improving agent performance.” → Could the authors point to the exact evidence to support this claim?

**Questions:**

1. Page 7: “... these critique data provide a more direct signal for π_W(θ) to improve its world modeling and planning ability during inference (see Section 5.3 for more quantitative study).” → Could the authors expand on why this is?
2. Section 5.1: “We suggest future work should scale both world model and policy training data together” → Isn’t that already happening in Figure 4? Since it says “After world model training, we perform one-round of policy training using the same set of policy learning data.”

---

> ### Author Response · Authors · 2025-11-17
> **Response to Weaknesses part 1**
>
> We thank reviewer G1k1 for acknowledging our “timely” work that aims to combine language agents with world models, robust evaluation including both in-domain and out-of-domain performance!
>
> ---
>
> > I’m not convinced that the benefits seen from doing imitation learning on R1 thinking reconstructions have anything to do with world modeling simulation. Specifically, did the authors try simpler prompts than the one listed in Table A4, maybe just keeping “remove unnecessary thinking parts without touching anything else”.
>
> In our preliminary experiments, we did explore other naive alternatives for DIT. For the simplified prompt you suggested, we asked GPT-4o to “remove any thinking segments unnecessary for deducing the final action” while keeping the final action untouched (denoted as “- simpler inst.” below). We also tried removing all thinking content (denoted as “- no-thinking”). Our results show that prompting GPT-4o to focus on world modeling simulation (i.e., Table A4) was necessary to achieve high performance while keeping token generation low.
>
> | Method | Gen. Token | Avg Success Rate |
> | :--- | :--- | :--- |
> | DirectDistill(R1) | 1.6x-6.0x | 20.9±1.0 |
> | DIT(ours) | 1.1x-2.5x | **22.6±1.0** |
> | - simpler inst. | 0.8x-2.0x |  20.5±1.2  |
> | - no-thinking |    0.1x-0.5x | 17.1±2.1  |
>
> We added this discussion in Section 3.2 and Appendix D in our revision.
>
> \
> > Figure 2 could be improved in terms of clarity: what’s the difference between Dyna Training (b) and Dyna-Think Training (c). Also, it seems (a) is just showing a general RL diagram - it’s unclear to me how this helps explain the proposed method?
>
> Dyna Training (b) refers to the classic Dyna RL algorithm (L154-158) which trains two separate models: a policy $\pi(\theta)$ and a world model $W(\mu)$. The goal is to improve policy training by using additional simulated experience generated by $W(\mu)$. In contrast, Dyna-Think Training (c) trains only one model, $\pi_W(\theta)$, using objectives for both world modeling and policy training. We believe including world model training serves as a crucial component for improving the model's environment understanding and planning capability. In our Table 1 and Table 4, we showed that DDT achieved both higher performance and higher world model accuracy compared to RFT and standard Dyna training.
>
> We believe Figure 3 (a), (b), and (c) illustrate very different paradigms in RL. (a) denotes the common setting with *only policy training* via interaction with the real environment. (b) additionally augments policy training with simulated experience, but requires training two separate models. Our method (c) combines both world model training and policy training from (b) into a single model. In general, we believe our Dyna-Think method presents a new way to view "Dyna" methods in the era of large reasoning models. However, we are open to discussion if this name is confusing.
>
> \
> > Page 5: “then constructing a policy learning dataset that trains π_W(θ) to predict each action a_i given the previous context context (o0,a<i,o<i) based on a reward function (e.g., task success).” → Could the authors be more specific here? Predicting each action given the previous context sounds like behavioral cloning, but the authors also mention a reward so I’m not sure exactly what objective is being optimized?
>
> We believe there is a misunderstanding, as we did not use behavior cloning for policy learning. Behavior cloning typically relies on expert demonstrations, which is not available for OSWorld or WindowsAgentArena. In this work, we use rejection sampling optimized by the policy gradient formula, as shown in Equation 2.

---

> ### Author Response · Authors · 2025-11-17
> **Response to Weaknesses part 2**
>
> > The paper could also use some improvements in writing. There are several sentences that contain spelling or grammar errors.
>
> Thank you for your suggestion. We have used grammar checking tools and fixed various grammar errors. Please see our revised manuscript.
>
> \
> > Section 3.3: why does providing a critique generated by GPT-4o have anything to do with world modeling?
>
>
> Given a generated action $a$, the critique compares the world model simulations in $a$ with the previous and next state to check whether the simulation is correct (L242-244). An example world model simulation could be “After opening the terminal with Ctrl+Alt+T, type ‘cp dir2/hi.txt dir1/’ to copy ...”, and an example critique is “Wait, maybe we need to first ensure that ‘dir1’ exists in the current directory...”.
> By training on such critiques, the model $\pi$ explicitly learns what are correct/incorrect world model simulations during its reasoning. In Table 4, we explicitly showed that DDT(critic) *achieved a higher world modeling accuracy (38.9 -> 55.7)* and task performance (22.6 -> 26.6) compared to baselines.
>
> \
> > Section 3.3: a lot of the algorithm is described in words, and it’d greatly help the clarity of exposition if some of this was written out more formally.
>
> To complement our verbal description of DDT, we provided pseudocode in Algorithm 1 and exact formulas used for optimization in Appendix E.2. Due to space limitations we kept it in the appendix. We hope Algorithm 1 could help clarify your confusions.
>
> \
> > Page 7: “… and that focusing on/improving simulation ability is effective at improving agent performance.” → Could the authors point to the exact evidence to support this claim?
>
> Our methods, DIT and DDT, are specifically designed to improve world model simulation ability. The results in Tables 1 and 2 show that both DIT and DDT outperform their respective baselines. ***More explicitly***, in Table 4 we measured world model simulation ability against task performance across different models. Our findings are twofold:
>
> 1.  DDT significantly improved world model simulation accuracy, *from 38.9 to 55.7*.
> 2.  Average world model accuracy for each task shows a strong correlation with task success, with a minimum correlation of *r=0.32* across all models (L470-472).
>
> We believe these results support our claim that world model simulation ability is a crucial component for agent performance.

---

> ### Author Response · Authors · 2025-11-17
> **Response to Questions**
>
> > Page 7: “... these critique data provide a more direct signal for π_W(θ) to improve its world modeling and planning ability during inference (see Section 5.3 for more quantitative study).” → Could the authors expand on why this is?
>
> Critique provides direct, verbalized feedback to the policy *during action generation*, indicating whether a simulation is good or bad. Since reasoning/action generation also happens during inference, this is more “direct”. On the other hand, methods like training models to predict the correct world model simulation is a separate task than action generation, and hence is more akin to multitask training to augment policy training. This is therefore more “implicit” compared to the critic approach.
>
> \
> > Section 5.1: “We suggest future work should scale both world model and policy training data together” → Isn’t that already happening in Figure 4? Since it says “After world model training, we perform one-round of policy training using the same set of policy learning data.”
>
> Figure 4 isolates the effect of scaling world model training by keeping the policy learning dataset fixed. In our revision, we rephrased the caption in Figure 4 for more clarity. We scaled world modeling data because our paper focuses on improving world modeling capability to improve policy.
>
> As described in Section 5.1, scaling world model training is feasible because it does *not* require knowing whether a trajectory is correctly or incorrectly solved, since only (state, action, next state) triplets in the interaction data are needed. This is in contrast to policy learning data, which often need evaluators to provide reward signals/distinguish between correct and incorrect trajectories. These evaluators are often written by humans and are challenging to be automated at scale.

---

> ### Author Response · Authors · 2025-11-24
>
> Dear reviewer G1k1,
>
> Thank you again for these questions and feedback! Please let us know at your earliest convenience if you have any further questions or concerns, or would like to us conduct any additional experiments.

---

### Official Review · Reviewer_f5TZ · 2025-11-01

**Soundness:** 3
**Presentation:** 3
**Contribution:** 2
**Rating:** 6
**Confidence:** 4

**Summary:**

The paper proposes Dyna-Think, a framework that integrates action-centric world-model simulation into an agent’s thinking process and then jointly improves the policy and its internal world model: (1) DIT reconstructs expert R1 thinking to keep only reasoning plus the world simulation directly relevant to the chosen action, and trains a policy on this curated data; (2) DDT is a two-stage, Dyna-style procedure that first trains internal world-model capabilities (via next-state prediction, state-difference, or critique prediction) and then improves policy via rollouts and rejection sampling. On OSWorld and WindowsAgentArena, DDT outperforms RFT and a vanilla Dyna baseline, and achieves similar Best-of-N success to R1 with ~2× fewer tokens.

**Strengths:**

- this work is generally well-written and presented cleanly
- the idea of integrating world model simulation into the cot process is interesting, and various variants of such an integration have been explored and compared
- the experiments are extensive, with useful ablations and analyses

**Weaknesses:**

- Why DDT > vanilla Dyna is under-explained. The authors show DDT (especially next-state and critique prediction) > vanilla Dyna under the same rollout budget (Table 1), but do not disentangle why world modelling helps in DDT and not in vanilla Dyna. Can the authors specify how the "separate $W(\mu)$" is trained/used in vanilla Dyna (e.g., model size, training signals, fidelity checks, rollout usage)?
- The authors say added evaluation hints are removed during training and testing in Sec 5.2, so what exactly are the hints used for? From Fig. 5a/b, it appears hints are appended only to rejection-sampling prompts to collect better policy data (w/ eval hint), not learned directly. Can authors clarify whether this is an ablation on critique-style training or a separate data-collection trick?

- for critique prediction, why does masking to critique tokens constitute “world-model training”? Is the critique treated as a predictive explanation of state difference? If so, any evidence that models trained with critiques actually improve next-state prediction accuracy (not only policy success)?
- also it is not clear what the DIRECT DISTILL is used for in Sec. 4.4?

- Given non-trivial data curation (rejection sampling, synthetic task generation, critique extraction), can the author add (or open source its code) a clear table listing, per method: number of tasks, real vs simulated rollouts used for policy vs world model training for both proposed method and baselines. This will help others replicate the presented results.

tiny presentation issues
- for easier scanning, it is recommended to bold the best proprietary model results in Table 1 and Table 3 too (you currently bold your methods only)
- The green/orange highlighting is hard to verify. Provide an explicit annotation protocol (what marks “simulation”), and an example with line-by-line tags so readers can see where simulation starts.

**Questions:**

- in Sec. 5.2, are evaluation hints used only to re-collect better trajectories via rejection sampling (and then removed), or do they leak into training? What is the precise purpose of this section—data bootstrapping ablation or critique-objective ablation?
- DIRECT DISTILL vs DIT (in Sec. 4.4): can the authors detail the pipelines? How is “R1 no-think” built, and how does DIT’s filtering to action-centric simulation differ from naive distill on all R1 tokens?

- why choose rejection sampling over GRPO/PPO-style RL for policy improvement? Did you try GRPO with the same budgets?

- sec. 5.1 scaling world-model training: which world modeling objective(s) were used? Do gains persist if the synthetic tasks come from a different generator or domain?

- what happens if you skip DIT and run DDT from the base Qwen2.5-32B-Instruct? This ablation would clarify DIT’s necessity vs DDT’s sufficiency.

- consider DIT reconstruction and critique injection both rely on GPT-4o for filtering/extraction and generating critiques, how do authors handle failure cases, and how sensitive is the judge? If we can switch to a stronger judge for a better performance?

---

> ### Author Response · Authors · 2025-11-17
> **Response to Weaknesses**
>
> We thank reviewer f5TZ for acknowledging our “well-written” work, “interesting” idea of integrating world model simulation into reasoning, and “extensive” experimental results!
>
> ---
>
> > Why DDT > vanilla Dyna is under-explained. The authors show DDT (especially next-state and critique prediction) > vanilla Dyna under the same rollout budget (Table 1), but do not disentangle why world modelling helps in DDT and not in vanilla Dyna
>
> Vanilla Dyna trains a separate world model to generate simulated trajectories for policy learning, which requires *predicting the full next screen state* at every step. This is an extremely difficult task in complex, long-horizon environments like OSWorld.
> In contrast, DDT (ours) avoids full state prediction and instead trains the model to *produce (partial) textual descriptions of future states within its reasoning process* (see the lower part of Figure 1 as an example). As shown in Table 1, enhancing world-modeling ability directly through reasoning is more tractable and more effective than vanilla Dyna.
> We added this explanation in our revised Section 4.3.
>
> \
> > The authors say added evaluation hints are removed during training and testing in Sec 5.2, so what exactly are the hints used for?
>
> During data collection, we include evaluation hints in the prompt so the model can solve more tasks using this additional guidance (similar to STaR; see L431). Figure 5(a) shows an example, where the evaluation configuration is appended after the instruction.
> After collecting trajectories, we remove these hints from the prompts during training *but keep the model’s generated responses unchanged*. This ensures a fair test-time comparison with baselines that never include hints in the prompts.
>
> \
> > for critique prediction, why does masking to critique tokens constitute “world-model training”? Is the critique treated as a predictive explanation of state difference? If so, any evidence that models trained with critiques actually improve next-state prediction accuracy (not only policy success)?
>
> Given a generated response a, the critique checks whether a simulation in a is correct or not based on the next state in the trajectory. An example simulation in reasoning could be “After opening the terminal with Ctrl+Alt+T, type ‘cp dir2/hi.txt dir1/’ to copy ...”, and an example generated critique is “Wait, maybe we need to first ensure that ‘dir1’ exists in the current directory...”. (L243-245)
>
> By training on this critique, DDT(critic) improves the model’s world-modeling ability, reflected in both higher task performance (Table 1) and *higher world-model accuracy (Table 4).*
>
> \
> > also it is not clear what the DIRECT DISTILL is used for in Sec. 4.4?
>
> In Section 4.4 we want to investigate the effect of different thinking patterns on performance, comparing them to simulation-focused reasoning in this work (DIT). To this end, we compared against (1) “fast-reasoning’’ traces from GPT-4o that lack simulation (Direct Distill-4o); (2) “full-reasoning’’ traces from R1 that include many cognitive behaviors beyond simulation (Direct Distill-R1); and (3) “no-thinking’’ traces created by removing R1’s thinking tokens (Direct Distill-R1 no-think).
> Table 2 shows that DIT achieves the best performance compared to the baselines. This suggests that the ability to perform reasoning with world-model simulations is indeed a central part in effective reasoning for long-horizon AI agent tasks.
>
> \
> > Given non-trivial data curation (rejection sampling, synthetic task generation, critique extraction), can the author add (or open source its code) a clear table listing, per method: number of tasks, real vs simulated rollouts used for policy vs world model training for both proposed method and baselines. This will help others replicate the presented results.
>
> In Algorithm 1, we provided the entire pseudo code for DDT training. In Table 1, we *did list the number of rollouts used for each method for both policy and world model training* (under the column $| \pi | $ and $|W|$, respectively), and also mentioned this briefly on L312-314.
> We are committed to open-sourcing our code and data once this work is accepted, so that our results can be replicated.

---

> ### Author Response · Authors · 2025-11-17
> **Response to Questions**
>
> > in Sec. 5.2, are evaluation hints used only to re-collect better trajectories via rejection sampling (and then removed), or do they leak into training? What is the precise purpose of this section—data bootstrapping ablation or critique-objective ablation?
>
> Yes, the evaluation hints are only used to collect better trajectories (compared to w/o evaluation hints). In order to provide a fair comparison with baselines during testing (which does not use evaluation hints), after collecting these trajectories we removed these hints from the prompts during training (see L449-550). During testing, *no methods have access to evaluation hints*.
>
> \
> > DIRECT DISTILL vs DIT (in Sec. 4.4): can the authors detail the pipelines? How is “R1 no-think” built, and how does DIT’s filtering to action-centric simulation differ from naive distill on all R1 tokens?
>
> For “R1 no-think”, we remove all text inside the “<think>…</think>” tags (see L377), so effectively each response in training only contains the action text. For DirectDistill(R1), we directly train on all tokens generated by R1.
> For DIT(R1), we reconstruct the reasoning traces to focus on world simulation, removing ~50% of R1’s tokens related to cognitive behaviors such as goal decomposition or self-reflection. Figure 1 illustrates this contrast by comparing R1’s original thinking process with our reconstructed version.
>
>
> \
> > what happens if you skip DIT and run DDT from the base Qwen2.5-32B-Instruct? This ablation would clarify DIT’s necessity vs DDT’s sufficiency.
>
> For DDT, we assume the base model already has some ability to perform world-model simulation during reasoning (e.g., so DDT(critic) has meaningful simulation text to critique). However, many non-reasoning models, such as Qwen2.5-32B-Instruct, lack this ability and hence requires a cold-start stage (e.g., DIT) before applying DDT.
>
> In general, we believe our method (DIT followed by DDT) mirrors common RLVR reasoning training setups, where a model is first trained to acquire strong reasoning ability before the final RL stage.
>
> \
> > consider DIT reconstruction and critique injection both rely on GPT-4o for filtering/extraction and generating critiques, how do authors handle failure cases, and how sensitive is the judge? If we can switch to a stronger judge for a better performance?
>
> We used GPT-4o for most data processing due to its strong general capability and its widespread use in prior work [1-3]. For tasks such as filtering and extraction, we found GPT-4o sufficiently reliable, though we agree that using a stronger model could improve judgment quality. For instance, using models like GPT-5-high may yield better results, but this comes with higher time and monetary costs.
> As an example, we additionally ran DeepSeek-V3 for DIT and DDT(critic) and report those results below. We generally find performance aligned with (slightly improved) our GPT-4o results. We added this discussion in our Appendix E.
>
> | Method | OSWorld Success Rate |
> | :--- | :--- |
> | DIT(w/ GPT-4o) | 22.6 |
> | DIT(w/ DeepSeek-v3) | 22.8 |
> | DDT(critic w/ GPT-4o) | 26.6 |
> | DDT(critic w/ DeepSeek-v3) | 27.0 |
>
> ---
>
> References
>
> [1] Xu, Yiheng, et al. "Aguvis: Unified pure vision agents for autonomous gui interaction." arXiv preprint arXiv:2412.04454 (2024).
>
> [2] Xie, Tianbao, et al. "Scaling Computer-Use Grounding via User Interface Decomposition and Synthesis." arXiv preprint arXiv:2505.13227 (2025).
>
> [3] Yang, Yuhao, et al. "Aria-ui: Visual grounding for gui instructions." Findings of the Association for Computational Linguistics: ACL 2025. 2025.

---

### Author Response · Authors · 2025-11-18

We thank all reviewers for their helpful feedback and suggestions! In addition to our individual responses, we have uploaded a revision of our paper accordingly. All changes are highlighted in red for ease of reference.

Please let us know if there is any further questions, concerns, or clarifications needed, and we look forward to any further feedback.

---

### Author Response · Authors · 2025-12-03

Dear AC, SAC, PC, and reviewers,


We would like to express our sincere gratitude for the time and effort devoted to the review. We are also saddened that this review cycle was affected by the recent unexpected event. Below, we provide a concise summary of the main questions and concerns raised during the rebuttal period and how our responses addressed them. We have also updated the manuscript accordingly, with revisions marked in red.

More specifically:
- **Reviewer f5TZ** acknowledged our “well-written” work, “interesting” idea of integrating world model simulation into reasoning, and “extensive” experimental results. The reviewer raised a few clarification questions such as “Why DDT > vanilla Dyna”, “what exactly are evaluation hints used for”, and “why does masking to critique tokens constitute ‘world-model training'". We provided detailed clarifications for each in our response and updated the manuscript accordingly. The reviewer also requested dataset statistics for each method (“given the non-trivial data curation…”); we note that these are already reported in our Table 1 under the columns $|\pi|$ and $|W|$.
- **Reviewer G1k1** recognized the “timely” nature of our work in combining language agents with world models, and noted the robustness of our evaluation across in-domain and out-of-domain settings. The reviewer first questioned whether “simpler prompts than the one listed in Table A4 (our proposed DIT)” would suffice. Through additional experiments, we showed that prompting GPT-4o to focus on world-modeling simulation (Table A4) was essential for strong performance while keeping token usage low. Reviewer G1k1 also raised clarification points and suggestions – e.g., improving the clarity of Figure 2 and providing more detail on policy training. We addressed these in our response and updated the manuscript accordingly.
- **Reviewer dbiP** acknowledged our “solid evaluation setup” across both in-domain and out-of-domain tasks, as well as the “robust and comprehensive” evaluation and analysis. The reviewer raised two main concerns: (1) the drop from 30.3% in-domain success to 17.6% out-of-domain. We clarified that our OOD tasks are substantially harder than ID, and this decline is consistent across models – for example, the untrained R1 also drops by 43% relative to its ID performance. (2) Whether our method depends on “proprietary GPT-4o” for data processing. We provided additional experiments showing that similar performance can be achieved using DeepSeek-V3. The reviewer also asked for clarification on comments such as “the ‘Dyna’ framing is evocative but confusing” and the need for deeper discussion on how next-state prediction may improve fundamental understanding. We responded to each in detail and believe these clarifications satisfactorily address the concerns raised.
- **Reviewer yrjn** acknowledged our clear motivation, “innovative” method, detailed experiments, and “well-written” method descriptions and results. The reviewer first asked whether “performance ceiling is primarily constrained by R1's performance limit, or is it limited by the scale of the base model”. We clarified that training models larger than 32B is beyond our computation resource, and we chose DeepSeek-R1 because, at the time of our work, it was among the strongest available reasoning models with demonstrated world-modeling ability. The reviewer then raised three additional points: (1) whether using GPT-4o introduces bias in the dataset or training process. We provided results showing that DeepSeek-V3 yields comparable performance. (2) what causes the “drop in OOD performance.” We explained that our OOD tasks are inherently harder than ID, and this decline is consistent across models – for example, the untrained R1 also shows a 43% relative drop. (3) whether we could break down the effect of synthetic data scaling on ID vs. OOD performance. We provided a detailed breakdown in our response.


Overall, we believe our responses address the core concerns and clarification questions raised by all reviewers. We hope the summary above helps the AC, SAC, and PCs more easily digest the key outcomes of the discussion period.


Sincerely, The Authors

---

### Meta-Review · Area_Chair_Aqt1 · 2026-01-05

**Summary:**

This work proposed to add "next state prediction and critique generation" (stated as world model task) into the objective of agentic LLM training. The technical component is the compression of trajectory tokens to action predictions. Using the state prediction subtask as an intermediate (like hidden layer) to regularize the model training. Basically, the idea especially the angle to research is somehow novel. However, as mentioned by reviewer dbiP "The current framing obscures the core contribution, which is about the structure of reasoning, not a classic learning-planning loop." The study of structure of reasoning, is not novel. Many efforts in literature (like Lookahead Search with MCTS) have been done to collect trajectory for agent training.
As two of the reviewers are negative. I think current rebuttal can not change their mind. Thus, I lean to reject.

**Reviewer Concerns:**

Below are points I think are not satisfied by reviewers:

1). G1k1: "I’m not convinced that the benefits seen from doing imitation learning on R1 thinking reconstructions have anything to do with world modeling simulation. Specifically, did the authors try simpler prompts than the one listed in Table A4, maybe just keeping “remove unnecessary thinking parts without touching anything else”.
The authors give additional results to compare the one with simple removing strategy:
DIT(ours)	1.1x-2.5x	22.6±1.0
- simpler inst.	0.8x-2.0x	20.5±1.2
However, the simpler inst.'s results is very comparable to DIT while using less tokens.
This indicates the proposed approach is not good enough even compared with very simple solutions.

2). G1k1: "Figure 2 could be improved in terms of clarity: what’s the difference between Dyna Training (b) and Dyna-Think Training (c). Also, it seems (a) is just showing a general RL diagram - it’s unclear to me how this helps explain the proposed method?"
The authors response that " In general, we believe our Dyna-Think method presents a new way to view "Dyna" methods in the era of large reasoning models. However, we are open to discussion if this name is confusing."
I think the term "Dyna" is misleading, the reviewer will not agree on this rebuttal.

3). Both reviewers argue on the use of gpt-4o: G1k1:"Section 3.3: why does providing a critique generated by GPT-4o have anything to do with world modeling?"
dbiP: "A key methodological concern is the framework's deep dependency on the proprietary GPT-4o model, which serves multiple roles as an editor, critic, and evaluator."
I agree with them. The work uses a lot of gpt-4o for compression to extract world model predictions.  This is not practicle as the performance need too many external strong model's efforts.

**Reviewer Scores:**

Both reviewer G1k1 and dbiP give negative scores. They are not included in the discussion. By reading the responses and reviews, I do not think the response would change their mind.

---

### Decision · Program_Chairs · 2026-01-26

Reject